# Cellular model system to dissect the isoform-selectivity of Akt inhibitors

Lena Quambusch [1,6], Laura Depta[1,6], Ina Landel[1], Melissa Lubeck[1], Tonia Kirschner[1], Jonas Nabert[1], Niklas Uhlenbrock[1], Jörn Weisner [1], Michael Kostka[2], Laura M. Levy[2], Carsten Schultz-Fademrecht[3], Franziska Glanemann[4,5], Kristina Althoff[4,5], Matthias P. Müller[1], Jens T. Siveke [4,5] & Daniel Rauh [1✉]

The protein kinase Akt plays a pivotal role in cellular processes. However, its isoforms' distinct functions have not been resolved to date, mainly due to the lack of suitable biochemical and cellular tools. Against this background, we present the development of an isoform-dependent Ba/F3 model system to translate biochemical results on isoform specificity to the cellular level. Our cellular model system complemented by protein X-ray crystallography and structure-based ligand design results in covalent-allosteric Akt inhibitors with unique selectivity profiles. In a first proof-of-concept, the developed molecules allow studies on isoform-selective effects of Akt inhibition in cancer cells. Thus, this study will pave the way to resolve isoform-selective roles in health and disease and foster the development of next-generation therapeutics with superior on-target properties.

[1] Faculty of Chemistry and Chemical Biology, TU Dortmund University and Drug Discovery Hub Dortmund (DDHD), Zentrum für Integrierte Wirkstoffforschung (ZIW), Dortmund, Germany. [2] Medicinal Chemistry, Taros Chemicals GmbH & Co. KG, Dortmund, Germany. [3] Lead Discovery Center GmbH, Dortmund, Germany. [4] Bridge Institute of Experimental Tumor Therapy, West German Cancer Center, University Medicine Essen, Essen, Germany. [5] Division of Solid Tumor Translational Oncology, German Cancer Research Center (DKFZ) and German Cancer Consortium (DKTK), partner site Essen, Heidelberg, Germany. [6] These authors contributed equally: Lena Quambusch, Laura Depta. ✉email: daniel.rauh@tu-dortmund.de

The protein kinase Akt is a central signaling molecule within the PI3K/Akt/mTOR-pathway, and dysregulation and malfunction of this pathway is a major cause of human diseases such as cancer and diabetes[1,2]. Biomarkers for those malfunctions are usually over-activated or altered upstream mediators[3]. Only rarely, overexpression of the Akt genes or the Akt[E17K] mutant resulting in elevated levels of active enzyme are reported to be causative for malignancies[4]. For the protein kinase Akt, three isoforms are encoded in the human genome sharing a high overall sequence homology, particularly within the conserved kinase and the pleckstrin homology (PH) domains. However, intracellular localization and tissue-specific expression levels seem to define their differing functionalities[5]. Akt1 is expressed ubiquitously and found in the cytosol as well as at the plasma membrane, whereas Akt2 can be found in muscle tissue and especially within mitochondria, while Akt3 is localized in the nucleus and shows high expression in neurons[6]. Invasive studies in knockout mice led to first insights into the individual isoform functions. Akt1 is linked to proliferation and antiapoptotic behavior[7]. Models with Akt2 deletions show a type-2 diabetic phenotype with impaired glucose uptake[8,9]. Absence of Akt3 is related to neuronal malfunctions and alterations in fatty acid metabolism[10,11]. However, similar genetic deletions in cancer models show opposing results and highlight the limitations of such approaches to dissect the actual roles and contributions of those isoforms within this proliferative disease. For example, knockout studies in aggressive breast cancer models suggest that Akt2 is responsible for metastasis, whereas in lung cancer models, Akt1 shows tumor-initiating behavior, thus entailing a different isoform-selective therapeutic strategy in the two examples[12,13]. To elucidate the functions of the three isoforms in less invasive perturbations highly selective bioactive small molecules are needed which can be used as molecular probes in concentration-dependent, and temporally controllable experiments[14]. Achieving the desired selectivity profile with small molecules is usually a complex and intricate task in pharmaceutical research, with the principal aim being to gain selectivity either for enzymes within a highly homologous family such as protein kinases or for selectively addressing disease-causing mutants while sparing their wild-type form[15,16]. The translation of those selectivity profiles into more complex cellular systems is a major bottleneck. Tissue-specific expression and different localizations of the isoforms within cellular compartments further complicate the development, evaluation, and direct comparison of isoform-selective inhibitors. Currently, straightforward model systems to assess isoform-selectivity on a cellular level are missing.

In this publication, we describe a cellular model system based on Akt-isoform-dependent Ba/F3 cell lines, which facilitated the evaluation of Akt isoform-selective inhibitors in a complex system. Furthermore, the cell-lines allowed direct comparison and translation of identified selectivity profiles from biochemical data to a cellular model system. Our recently developed pyrazinone-based covalent-allosteric Akt inhibitors (CAAIs)[17] inspired our structure-guided design approach, and combined with a thorough analysis of the Akt isoform homology models, we designed and synthesized a set of diverse and chemically accessible pyridine-based CAAIs to explore the allosteric binding pockets in more detail. Additional structural data of two full-length Akt1 crystal structures in complex with this innovative class of inhibitors confirmed our successful design approach. In addition, we used quantitative kinetic measurements to further characterize the synthesized compounds and show the covalent modification of Akt3. Thus, the combination of biochemical characterization, together with the structural data and the herein-developed cellular system, enabled us to identify at least one promising ligand for each Akt isoform. These ligands qualify as chemical probes for

further perturbation studies and showed promising results in the cancer cell line PANC1.

## Results

**Murine pro-B Ba/F3 cell line as a tool system**. Initial immunoblotting experiments with our previously reported isoform-selective inhibitors in genetically altered cellular models demonstrated a dose-dependent efficacy and on-target inhibition but did not allow for assessment of the inhibitors' anti-proliferative activity. Based on those findings, we were motivated to develop an efficient and robust cellular model intended to resolve isoform selectivity of promising inhibitors in a high throughput setting. We chose a prominent cellular model system in the field of kinase research, which is based on the murine pro-B Ba/F3 cell line, whose survival and proliferation are dependent on the exogenous supply of the growth factor interleukin-3 (IL-3)[18]. Additionally, previous studies implicated an important function of the Akt signaling pathway in IL-3-mediated survival by inhibiting the intrinsic apoptotic machinery of Ba/F3 cells (Fig. 1a)[19,20]. By integrating dominant oncogenes via retroviral transduction, the dependence of cell survival on the IL-3-induced signaling pathway can be transferred to the constitutive proliferation signal of the corresponding driver mutation, thereby achieving oncogenic addiction of Ba/F3 cells to the transgene[21]. This renders the IL-3 induced signaling pathway an ideal tool to test for the differential cytotoxicity of drug candidates or whole compound libraries in a high-throughput manner and thus represents a suitable comparative system to address the individual Akt isoforms[22].

**Generation of stable Akt isoform-dependent Ba/F3 cells**. To introduce constitutively active forms of the Akt isoforms (Akt1/Akt2/Akt3) into murine Ba/F3 cells, we used retroviral constructs, each carrying a myristoylated Akt isoform. This lipidation of Akt causes a preferential association with exogenous and endogenous membranes, leading to the constitutive phosphorylation and an increased fraction of activated kinase (Fig. 1b). The expression of the transgenes, and their influences on downstream signaling, were confirmed by western blot analysis. By measuring the binding of Akt isoform-specific antibodies, strong overexpression of the individual Akt isoform compared to the parental Ba/F3 cell line was detected. Further, the results show a significant increase in the amount of pAkt[S473], indicating a high level of active Akt isoform in the corresponding cell line. Additionally, the myr-Akt isoforms trigger an increased phosphorylation of the downstream targets FOXO and GSK-3β, while maintaining the phosphorylation level of PRAS40, indicating the activation of downstream signaling pathways. In contrast, pErk1/2, and the ribosomal protein S6, which are involved in the regulation of cell proliferation, show weaker phosphorylation than the parental cell line. To investigate the generated cell lines' growth behavior, we used the CellTiter-Glo® Luminescent Cell Viability Assay (Promega). The results show a slower growth rate for Ba/F3 myr-Akt1/2/3 compared to the parental Ba/F3 cell line, which might be caused by the decreased phosphorylation of ERK1/2 and S6 (Fig. 1c).

**Ba/F3 constructs are sensitive to targeted Akt inhibition**. In order to gain detailed insights into the molecular mode of action and its effects on Akt signaling, we tested the responsiveness of the transformed cell lines upon Akt inhibition by utilizing the previously reported CAAI borussertib and capivasertib, an orthosteric inhibitor[23]. Borussertib showed a dose-dependent reduction of pAkt, as well as the downstream targets pPRAS40, pFOXO, and pGSK3β, whereas the ATP-competitive inhibitor is less effective (Supplementary Fig. 1). Furthermore, higher inhibitor concentration led to increasing cPARP levels, which

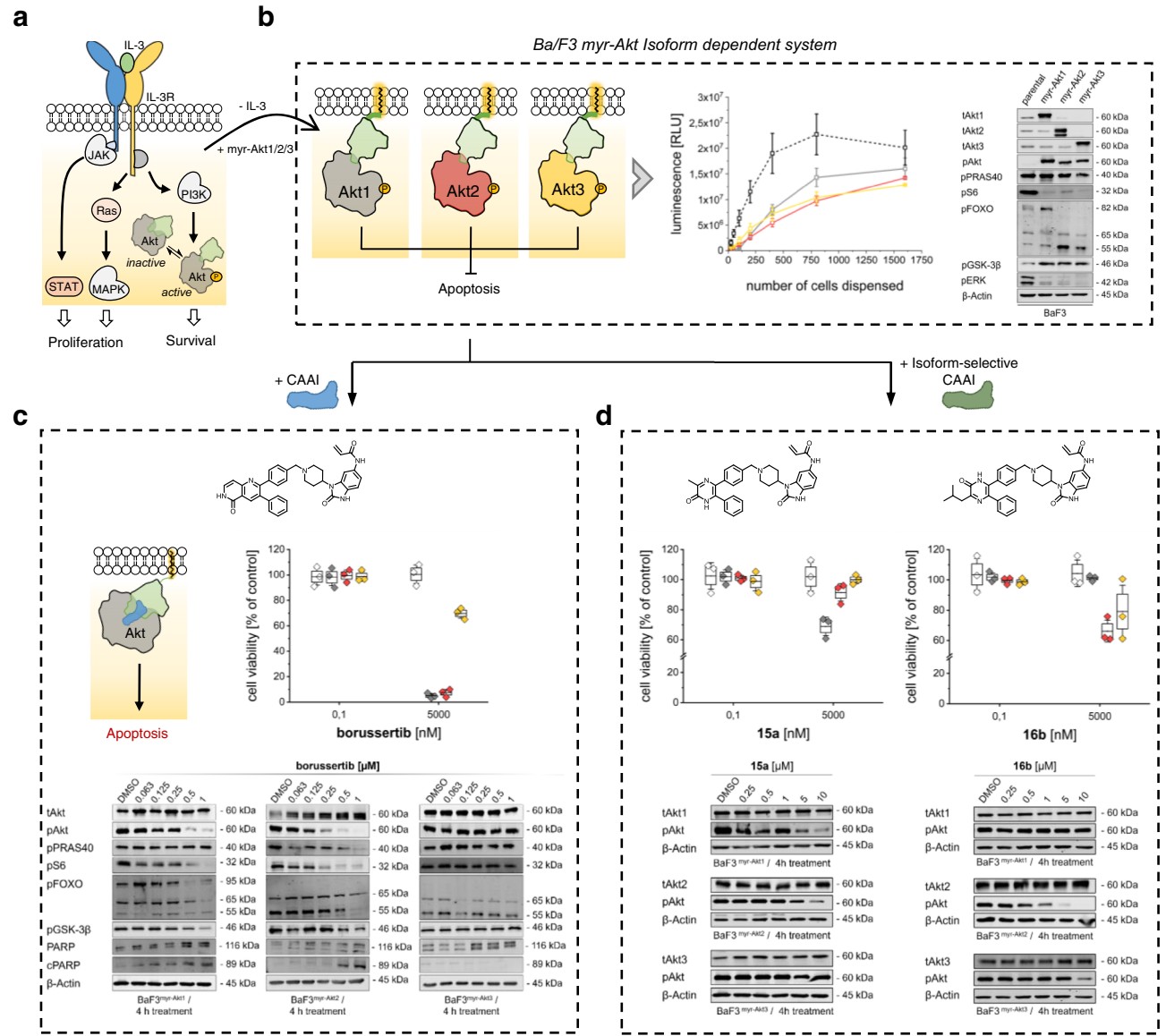

**Fig. 1 Generating Ba/F3 myr-Akt isoform-dependent cell lines to evaluate targeted Akt inhibition. a** Representation of the IL-3 signaling pathway in the murine Ba/F3 cell upon binding of IL-3. **b** General overview of the established cell lines Ba/F3$^{myr-Akt1}$ (gray), Ba/F3$^{myr-Akt2}$ (red), and Ba/F3$^{myr-Akt3}$ (yellow). Graph showing the seeded cell number of the Ba/F3 constructs and resulting luminescent signals following treatment with CTG solution compared to the parental system (dotted line, white). The experiments were performed in triplicates with $n = 3$. Error bars indicate s.d. Immunoblot of myr-Akt isoform systems highlighting the molecular expression levels, activities, and alterations of downstream targets of Akt ($n = 1$ biologically independent experiment). **c** Addressing the Ba/F3 myr-Akt with CAAI borussertib to investigate the sensitivity of the established system. Structure of used inhibitor and schematic representation of the binding mode. Graph showing the remaining viable cells as a percentage with respect to the used concentration of borussertib. Experimental points were measured in duplicates for each plate and were replicated in $n = 3$ biologically independent experiments. Immunoblots resolved individual Akt isoforms' presence and activity within the three model systems after 4-h treatment ($n = 1$ biologically independent experiment). **d** Addressing the Ba/F3 myr-Akt cells with isoform-selective CAAIs **15a**/**16b** to investigate selectivity translation into the established system. Graph showing the remaining viable cells as a percentage with respect to the used concentration of **15a** and **16b**. Experimental points were measured in duplicates for each plate and were replicated in $n = 3$ biologically independent experiments. Structure of used inhibitors. Immunoblots resolved isoform selective activity within the three model systems after 4-h treatment ($n = 1$ biologically independent experiment). In **c**, **d** the box plot middle line represents the mean value, the box represents the first and third quartile and the error bars indicate s.e. Source Data are provided with this paper.

confirmed the induction of apoptosis by Akt inhibition and demonstrated the myr-Akt isoform-dependency of the Ba/F3 cells (Fig. 1c). Cell viability studies revealed significantly reduced amounts of viable cells for Ba/F3$^{myr-Akt1}$ and Ba/F3$^{myr-Akt2}$ in the presence of increasing borussertib concentrations. Additionally, the Ba/F3$^{myr-Akt3}$ cell line's low response gave the first hint of the applicability of the generated cell lines to resolve targeted inhibition of the individual Akt isoforms as the inhibitor shows a

moderate activity (IC$_{50}$ = 500 nM) for Akt3 in biochemical assays (Fig. 1c), compared to 1 nM for Akt1 and 14 nM for Akt2 (see Table 1)[17]. In further experiments, we thus used our recently published pyrazinone-based isoform-selective CAAIs **15a** and **16b**, to evaluate the cellular model system, and to examine isoform-selectivity. Western blot studies with concentrations ranging from 10 μM to 250 nM displayed different sensitivities in the myr-Akt isoform-dependent cell lines (Fig. 1d and

**Table 1 Biochemical evaluation of covalent-allosteric Akt inhibitors with the Akt isoforms.**

| # | R | IC$_{50}$ [nM] | | |
|---|---|---|---|---|
| | | Akt1 | Akt2 | Akt3 |
| Capivasertib | | 1 ± 0.3 | 4 ± 1 | 8 ± 1 |
| MK-2206 | | 10 ± 2 | 54 ± 18 | 720 ± 121 |
| Borussertib | | 1 ± 0.2 | 14 ± 1 | 431 ± 31 |
| 1 | | 53 ± 13 | 599 ± 63 | 6627 ± 1128 |
| 2 | | 22 ± 2 | 150 ± 5 | 7509 ± 1821 |
| 3 | | 186 ± 79 | 961 ± 174 | >20,000 |
| 4 | | 44 ± 18 | 248 ± 49 | 6446 ± 742 |
| 5 | | 35 ± 4 | 116 ± 2 | 3396 ± 406 |
| 6 | | 112 ± 21 | 108 ± 31 | 5752 ± 1547 |
| 7 | | 7087 ± 3437 | 479 ± 91 | >20,000 |
| 8 | | 4566 ± 197 | 2068 ± 147 | >20,000 |
| 9 | | 5930 ± 793 | 4214 ± 412 | 8374 ± 1784 |
| 10 | | 1716 ± 166 | 94 ± 4 | 1564 ± 75 |
| 11 | | 643 ± 95 | 76 ± 11 | 1890 ± 621 |
| 12 | | 1281 ± 18 | 251 ± 18 | 607 ± 95 |
| 13 | | 98 ± 15 | 53 ± 5 | 3543 ± 617 |
| 14 | | 527 ± 14 | 23 ± 4 | 356 ± 88 |
| 15 | | 604 ± 31 | 1400 ± 112 | 4356 ± 1077 |
| 16 | | 9484 ± 665 | 682 ± 83 | 5339 ± 2006 |
| 17 | | 2853 ± 758 | 133 ± 9 | 2108 ± 441 |
| 18 | | 91 ± 13 | 39 ± 6 | 5266 ± 1030 |
| 19 | | 1797 ± 105 | 511 ± 62 | 1509 ± 389 |
| 20 | | 1631 ± 148 | 175 ± 2 | 187 ± 40 |
| 21 | | 2111 ± 268 | 99 ± 6 | 272 ± 84 |
| 22 | | 209 ± 40 | 55 ± 5 | 3049 ± 646 |
| 23 | | 130 ± 11 | 151 ± 25 | 6303 ± 1923 |

Data presented as mean values ± s.d; $n = 3$, where $n$ represents the number of independent experiments.

Supplementary Fig. 2). In particular, the 6′-modified ligand **15a** was more potent towards myr-Akt1 than myr-Akt2 and myr-Akt3, as indicated by a dose-dependent decrease of pAkt$^{S473}$ as well as a lower amount of viable cells detected in the Ba/F3$^{myr-Akt1}$ cell line after treatment. For myr-Akt2 and myr-Akt3, the data showed preferential binding of the 5′-substituted molecule **16b**. Altogether the results showed a very good correlation to the previously reported biochemical data and isoform-selectivity[17].

**Homology-model highlights variation in allosteric pocket.** Up to now, no full-length crystal structures of Akt2 and Akt3 have been reported. Thus, as an alternative, we used homology models for Akt2 and Akt3. Detailed and thorough sequence analysis of the Akt isoforms revealed significant variations within the allosteric binding sites[17,24]. Those findings suggest an altered helical structure in the interdomain pocket as a key difference between the isoforms. Homology models for Akt2 and Akt3 based on the new Akt1 co-crystal structures (PDB: 6S9X)[17] were built (Fig. 2b). In agreement with previous modeling results, these models exhibit a shortened c-terminal end of the αE-helix around residues 265–270. The model of Akt2 is in good agreement with a crystal structure of the inactive kinase domain (PDB: 1MRY, Supplementary Fig. 3)[25]. In Akt3, the polar Lys269 is more solvent exposed. In comparison to the Akt1 structure, a larger accessible space at the end of this interdomain pocket is formed, which might allow incorporation of larger moieties into inhibitors for Akt2 and especially Akt3 (Supplementary Fig. 4). This is in agreement with previously published findings, where the introduction of bigger moieties at the 5′-position in the pyrazinone-based CAAIs was tolerated by Akt2 and Akt3, but not Akt1 (Fig. 2c). According to the model each enzyme has a different amino acid with distinct chemical properties at position 269, after the c-terminal end of the αE-helix: an asparagine in Akt1, an aspartate in Akt2, and a lysine in Akt3, thus introducing significant changes in the electrostatic potential at this position in the different isoforms (Supplementary Fig. 5). To verify and validate the aforementioned contributions to different putative selectivities between the isoforms, we undertook a detailed study of the structure-activity relationships (SAR) with a focused design of a diverse set of small molecules.

**Structure-guided design to explore allosteric pocket.** Recently, we utilized a class of covalent-allosteric Akt inhibitors (CAAIs) to achieve isoform-selectivity. These types of ligands alkylate one of two cysteine residues within an interdomain pocket between the regulatory PH and the catalytic kinase domain, leading to an irreversible stabilization of the inactive conformation bearing a structurally blocked ATP-binding site (Fig. 2a)[23,26]. The modeling results guided our drug-design approach toward a focused library of covalent modifiers. Here, the focus was on molecules with a 5′-position substitution, a modification which was previously shown to be favored by Akt2 and Akt3. Variations in chemical properties and spatial demand were the prime objective for the introduced moieties (Fig. 2d). Those substitutions should be connected through different linkers to the main biarylic pyridine-core scaffold. The similarity to the formerly used pyrazinones is relatively high and preserves the important structural elements for binding: two π–π driven stacking interactions: with Trp80 and Tyr272. Extensive research in this field already showed the importance of nitrogen within the Trp80-stacking ring system[27,28]. Thus, the compound series shown in Table 1 was synthesized and used to further evaluate the structure-activity relationship toward the allosteric binding sites in the Akt isoforms.

**Chemically accessible pyridines as scaffold for derivatization.** To prepare this diverse set of molecules, we developed an efficient and divergent synthesis route with late-stage functionalization (Supplementary Fig. 6 and Supplementary Methods). Based on a trisubstituted pyridine-core scaffold, this synthetic strategy enabled a simple route for the desired diversification of the planned covalent-allosteric inhibitors. Starting with the conserved linker and warhead bearing-part of the resulting ligands **S5** equipped with a boronic acid for further modification. Therefore, a benzo[d]imidazolone was transformed to molecule **S1** via a selective *ortho*-nitration[29]. Subsequent Boc-protection of the secondary amine followed by a Pd-catalyzed reduction of the introduced nitro-group and a dehalogenation led to molecule **S3**[30]. Then the attachment of the acrylamide followed by acidic deprotection gave the intermediate **S4**, which was transformed into the desired building block **S5** via reductive amination in presence of formyl phenylboronic acid[31]. The more variable and diverse part of the CAAI's were synthesized starting from the corresponding dihalogenated pyridine or nicotinic building blocks. In all variants the first step involved a chemoselective Suzuki-Miyaura reaction, introducing a phenyl moiety at 3′-position (**S8–S11**)[32]. The Methylpyridine **S9** was selectively brominated at the benzylic position (**S12**), which allowed in the following an introduction of diverse moieties via nucleophilic substitution or Suzuki cross-coupling at this position to yield the derivatives **S13–S22**[33]. On the other hand, the nicotinic ester derivative **S10** was deprotected under basic conditions and then diversified via amide coupling to give a variety of building blocks (**S24–S30**)[34]. In a final step the warhead bearing linker molecule **S5** was connected to the diverse set of pyridine-based building blocks through a microwave-assisted Suzuki cross-coupling to give the desired CAAI's (**1–23**). In total, 23 inhibitors were synthesized, exemplifying a focused library with distinct modifications and chemical properties to investigate the structure-activity relationship of the Akt isoform interdomain binding pockets.

**Biochemical activities outline selectivity profiles.** To gain proper insights into the potencies and selectivities of the ligands, the inhibitory potency (IC$_{50}$) towards the three Akt isoforms was evaluated in a biochemical activity-based assay (Table 1).

In general, the plain pyridine scaffold (**1**) was tolerated and showed a good affinity for Akt1 and a 10-fold and 100-fold loss of activity for Akt2 and Akt3, respectively. Overall, the introduction of larger moieties was less tolerated by Akt1. In contrast, smaller groups like the free carboxylic acid (**4**) or corresponding ester (**5**), as well as a methyl group (**2**) at the pyridine core, increased the potency. Residues which can be protonated and possess a positive charge under physiological conditions, e.g., the pyrrolidine (**7**), were disfavored. The linkage of some residues by an amide bond seemed to be more tolerated in direct comparison to the methylene-bridged version (**9** vs. **10**, **17** vs. **18**, **21** vs. **22**), whereas the simple methyl amide (**6**) lost 4-fold activity compared to the ester version. The nitrogen's less basic character might be beneficial for Akt1 as well, and this structural motif has been reported in CAAIs before[23]. In the case of the Akt2 interdomain binding site, larger residues were more favored than smaller moieties. Basic secondary amines, as in pyrrolidine moieties, were tolerated with moderate activity, whereas the amide-linked version of those ligands showed a 5-fold drop in affinity. (**7** vs. **8**). Small aromatic ring systems such as methyl pyrazole (**12/13**) and imidazole (**11**) showed good activities (IC$_{50}$ = 53–251 nM). The nitrogen in those aromatic rings was especially favored compared to the oxygen-bearing system (**14** vs. **15**). A similar effect was observed within the six-membered electron-rich systems, especially 3-pyridine (**21**) and

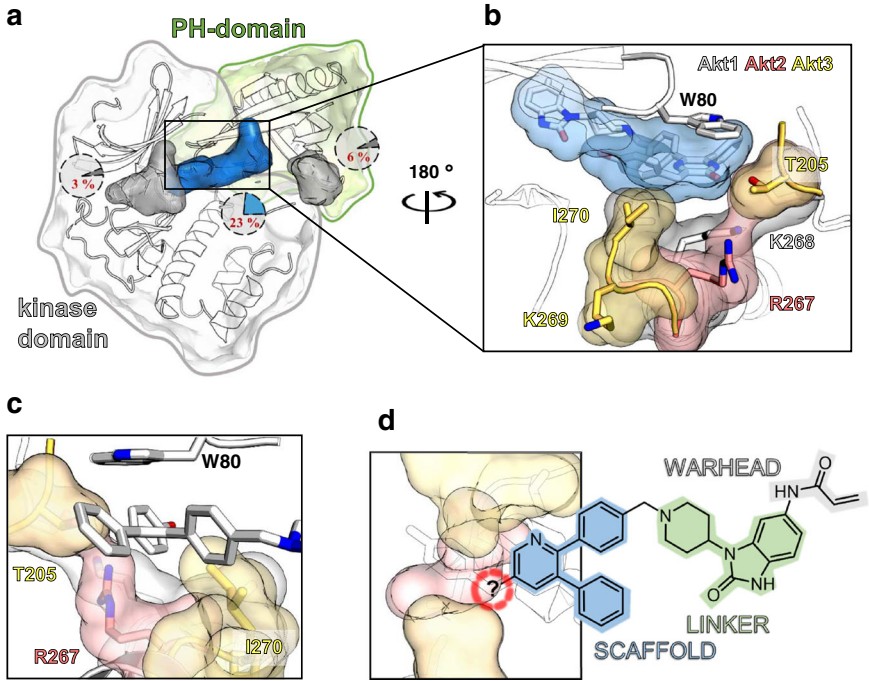

**Fig. 2 Homology-model guides design of covalent-allosteric inhibitors to address Akt isoforms. a** General overview of the closed conformation of protein kinase Akt (kinase domain: gray, PH domain: green; based on PDB: 6HHF)[36] and the druggable binding pockets (ATP-pocket within kinase domain: gray (PDB: 4GV1)[52], allosteric-interdomain pocket: blue (PDB: 6HHF), PIP$_3$-binding pocket within PH domain: gray (PDB 2UVM)[53]). The variety of the binding site forming residues among the three isoforms are given in percentage; **b** View of the interdomain binding site in Akt1 (white; PDB 6S9X: with the covalent-allosteric Akt inhibitor **15c** (blue))[17] in comparison to homology models of Akt2 (red) and Akt3 (yellow). The structural changes within the allosteric site and the different side chain residues of each isoform are highlighted; **c** Modeling studies with recently found pyrazinonic CAAI **16c** into Akt isoform homology model highlight structural differences and possible interactions within the allosteric binding site (Akt1(white) PDB: 6S9X; Akt2 (red) and Akt3 (yellow) homology model)[17]. The side view shows an enlarged space in Akt2 and Akt3, ideally to allow the binding of the hydroxyl phenyl moiety (Supplementary Fig. 4); **d** Structure-guided design approach towards a focused set of small molecules for evaluation of Akt isoform-selectivity: trisubstituted, biarylic pyridines function as core scaffold (blue). A benzimidazolone linker (green) is decorated with the Michael acceptor warhead (gray) to address the nucleophilic cysteine residues within the allosteric binding site. Different moieties (red) with various chemical properties and spatial demand should be connected over a defined linkage to the core molecule.

aminopyridine (**20**) showed a good affinity towards the Akt2 binding pocket. Further, the simple aniline group (**17**) was more potent than the phenol (**16**), whereas modifications at the 4-position of those rings were less well tolerated (**19**). For the third Isoform Akt3, the depicted set of CAAIs was overall less active, and only a few compounds showed a moderate potency. Smaller substituents were not favored, and bulkier moieties were needed to inactivate the protein. The aminopyridine (**20**) and pyridine (**21**) as well as the methyl pyrazole (**12**) showed good sub-micromolar activity. Again, the importance of the nitrogen within the aromatic ring system was emphasized by the anilinic CAAI (**19**), which showed a 10-fold loss of potency. Thus, both the size of the corresponding pocket as well as the importance of the different surface polarization within the targeted area of the individual binding pockets are consistent with the results presented here. Especially Akt2 tolerates positively charged residues in pyrrolidine (**7**) and pyrrole (**14**), which is presumably caused by favorable electrostatic interaction with Asp269. Interestingly, Akt2 seems to tolerate both linkages, amide or methylene, whereas Akt1 prefers the amide-linked version and Akt3 the methyl ones, e.g., pyridine 21 and 22. In agreement with the observed changes at the end of the αE-helix mentioned above (Supplementary Fig. 4), Akt2 shows a hybrid-like character between Akt1 and Akt3 regarding the SAR of this inhibitor series.

**Kinetics and Protein MS resolve covalent modification.** For ligands that showed a promising activity towards Akt1, Akt2, or Akt3 (IC$_{50}$ < 500 nM), an additional detailed analysis was performed to obtain the inhibitory binding constant $K_i$, the rate constant of covalent bond formation $k_{inact}$ and the second order rate constant $k_{inact}/K_i$ (Table 2).

The kinetic data indicated that the tested inhibitors possess a moderate inactivation rate on all three isoforms. In general, the resulting $k_{inact}/K_i$ values, i.e. the efficacy of covalent complex formation, are in good agreement with the observed potencies of the evaluated inhibitor set as well as the identified selectivity ratios. Compared to the well-described inhibitor borussertib, most ligands have slightly weaker affinities on Akt1 and Akt2. Notably, the inactivation rates are slightly higher on Akt3 than on the other isoforms, especially for CAAI **20**. However, the reactivity of the two cysteine residues that are known to be addressed in Akt1 have not been thoroughly characterized in Akt2 and Akt3 yet and thus require further investigation.

Protein mass spectrometry analysis provided orthogonal evidence for the covalent bond formation. The set of CAAIs was evaluated with the three Akt isoforms. All mass spectra showed a mass shift corresponding to the mono-labeled protein as compared to the apo protein (Supplementary Fig. 8). The targeted covalent modification of Akt3 was just described recently by Aye et al. for Cys119 within the linker region and is shown here at Cys293[35]. Further MS/MS experiments revealed selective labeling with our covalent inhibitors of the targeted cysteine residues in the activation loop, as expected from previous results with Akt1 (Supplementary Fig. 9–11).

**Akt1 co-crystal structures reveals binding mode**. For a greater understanding of the binding mode of the synthesized ligands, we co-crystallized two Akt1-selective pyridine derivatives with full-length Akt1 (Fig. 3). The complex structures depict a similar binding mode as reported before and show that both ligands stabilize the full-length protein in the autoinhibited conformation with the PH domain folded onto the kinase domain (PH-in conformation)[26,36]. Through this intramolecular contact between the N- and the C-lobe, the regulatory helix αC is displaced, shaping an allosteric binding pocket at the interface. The kinase is stabilized in a DFG-out conformation. Due to missing interactions with adenosine or an equivalent moiety, the hinge region is slightly shifted. The relatively small biarylic part of the molecule is stabilized by π–π-stacking with Trp80 and Tyr272. Additional hydrophobic interactions can be observed between the phenyl ring in the 3-position and Leu210, Leu264, and Ile290. The structure in complex with **6** showed that the methylamide attached to the main scaffold had a planar position, which bypassed a potential steric clash with Lys268. On the other hand, the complex with CAAI **3** highlighted a more flexible residue at the benzylic position of the pyridine core, which is close to Lys268. In both structures, the data suggested that the covalent inhibitors can label Cys310 and Cys296 (Supplementary Figs. 12 and 13). Further crystallization studies with full-length Akt2 and Akt3 are a necessity to extend our structural understanding and finally provide detailed insights into the interactions of the binding pockets of the three isoforms.

**Cellular Ba/F3 model system reveals promising inhibitors**. To confirm the applicability of the Ba/F3 myr-Akt isoform-dependent system to test differential isoform-selectivity, we utilized the previously described focused inhibitor set.

To evaluate the translation of the selectivity profiles, we compared the CTG EC$_{50}$ selectivity log ratios Akt1/Akt2 and Akt2/Akt3 with the corresponding HTRF IC$_{50}$ log ratios (Fig. 4a and Supplementary Table 3). Ligands equipped with polar groups, e.g., free acid or hydroxyl group (**3**, **4**, **16**), were generally inactive on the Ba/F3 myr-Akt isoforms cell lines, which is presumably caused by low cellular permeability (Supplementary Table 2). However, in general, we observed that the selectivity ratios within the cells related well to the in vitro trends for the analyzed Akt-isoform-selective inhibitor set, underlining the applicability of our engineered cellular system to resolve isoform selectivities. The results showed that molecules **11**, **14**, and **17** maintained their selectivity ratios for myr-Akt2 in a cellular context, confirming that bigger residues were more favored by the Akt2 binding pocket than smaller moieties. In agreement with the biochemical data, the aminopyridine (**20**) and pyridine (**21**) as well as the methyl pyrazole (**12**), showed slightly shifted activities towards the isoform myr-Akt3 equipotent to myr-Akt2 underscoring the importance of the nitrogen within the aromatic ring system for targeting Akt3. The plain pyridine scaffold (**1**) shows the expected moderate selectivity for myr-Akt1 as well as a loss of activity as the attached residue increased in size. Due to their promising selectivity profiles, which were also confirmed via western blot studies (Fig. 4b and Supplementary Fig. 14), we chose the CAAI's **1**, **14**, and **20** for further investigations.

**Isoform-selective ligands show on-target effect in PANC1**. We next examined the on-target inhibition of molecule **1**, **14**, and **20** in the human, pancreas-derived cancer lineage PANC1 as a proof-of-concept and used PANC1 cells, whether harboring an Akt1 knockout or an Akt2 knockout, as a cross validation for inhibitor induced decrease of downstream phosphorylation signals (Supplementary Fig. 15). We employed a cellular thermal shift assay in combination with immunoblotting experiments (Fig. 5 and Supplementary Fig. 16). Treatment of the PANC1 cells with **1** revealed a downregulation of pAkt1 and pAkt2, indicating that **1** is not able to maintain its selectivity in this cell line. In contrast, treatment with **14** results in efficient downregulation of Akt2 phosphorylation and its downstream targets PRAS40, consistent with the observed pPRAS40 downregulation in the PANC1 knockout models, and S6, while the pAkt1 signal remained stable. In line with the previously reported data, treatment with molecule **20** induced a moderate downregulation of pAkt2 (Fig. 5a). Due to its low phosphorylation level in PANC1, Akt3 phosphorylation could not be resolved via immunoblotting, but an on-target stabilization was validated through CETSA experiments. While a treatment with pan Akt inhibitor borussertib induced a stabilization of all three Akt isoforms, we observed a significant thermal stabilization of Akt2 in comparison with the DMSO control as well as Akt3 and Akt1 following a treatment with **14** (Fig. 5b and Supplementary Fig. 17a). Furthermore, the results point to a slight stabilization of Akt3 using 1 μM of molecule **20** (Fig. 5b) and a significant stabilization of Akt2 and Akt3 using 2.5 μM of **20** (Supplementary Fig. 17), indicating the equipotency of **20** on these isoforms. By using 0.3 μM of **1**, we were able to show a preferred stabilization of Akt1, while losing this selectivity when the concentration of **1** is increased to 1 μM (Fig. 5b and Supplementary Fig. 17b).

## Discussion

Inconclusive results of Akt isoform-selective targeting within cellular model systems have made a proper evaluation and translation of biochemical results of isoform-specific CAAIs difficult and tedious in the past, presumably because of the different functions, expression levels, and localization of the Akt isoforms. Therefore, we set out to establish a comparative cellular model system by generating myr-Akt isoform-dependent Ba/F3 cells using retroviral constructs, each harboring a single myr-istoylated Akt isoform. We were able to detect the presence of an increased fraction of active Akt isoform in the corresponding transduced cell line and the ability of the myr-Akt constructs to activate the downstream targets FOXO and GSK3β. Overall, it has to be noted that this is an artificial cellular system derived from a murine origin. For this reason, we believe that any conclusions about downstream effects need to be viewed with the necessary caution. However, due to the low amount of other active Akt isoforms, this cellular model represents an ideal tool to test for selectivity profiles of CAAIs and their transferability into a cellular context.

Additionally, and as a proof-of-concept, we utilized and designed a set of covalent-allosteric inhibitors to specifically address the protein kinase Akt isoforms and tested these in a previously established biochemical assay. Subsequently, we also showed that the biochemical data was readily transferable to the cellular system established in this study. Our design was based on homology models for Akt2 and Akt3, which underline major structural differences in the allosteric pocket. The synthesized set of covalent-allosteric inhibitors had varying chemical properties in the introduced residues at the 5′-position of the pyridine-core scaffold. The data suggest that the amide linker is essential to obtain potent inhibitors for Akt1. In some cases, the rigid and more planar linker rescues the activity up to a 10-fold increase. On the other hand, Akt3 shows the exact opposite behavior and the CAAIs lose affinity upon introduction of the amide, suggesting that flexibility is crucial for a beneficial interaction. In Akt2, positively charged moieties seem to be better tolerated than negative ones, thus highlighting the influence of the amino acid (Asp269) in that area of the binding pocket.

**Table 2 Kinetic evaluation of covalent-allosteric Akt ligands with the three Akt isoforms.**

| # | Akt1 | | | Akt2 | | | Akt3 | | |
|---|---|---|---|---|---|---|---|---|---|
| | $K_i$ [nM] | $k_{inact}$ [$10^3$ min$^{-1}$] | $k_{inact}/K_i$ [mM$^{-1}$s$^{-1}$] | $K_i$ [nM] | $k_{inact}$ [$10^3$ min$^{-1}$] | $k_{inact}/K_i$ [mM$^{-1}$s$^{-1}$] | $K_i$ [nM] | $k_{inact}$ [$10^3$ min$^{-1}$] | $k_{inact}/K_i$ [mM$^{-1}$s$^{-1}$] |
| 1 | 6 ± 2 | 12 ± 3 | 38 ± 10 | – | – | – | – | – | – |
| 10 | – | – | – | 4 ± 1 | 6 ± 1 | 25 ± 5 | – | – | – |
| 11 | 9 ± 2 | 12 ± 2 | 26 ± 2 | 11 ± 1 | 12 ± 2 | 20 ± 4 | – | – | – |
| 13 | 8 ± 2 | 14 ± 1 | 31 ± 11 | 40 ± 3 | 19 ± 1 | 8 ± 1 | – | – | – |
| 14 | – | – | – | 11 ± 2 | 16 ± 3 | 25 ± 6 | 29 ± 4 | 22 ± 2 | 13 ± 2 |
| 18 | 6 ± 1 | 10 ± 2 | 32 ± 5 | 11 ± 2 | 10 ± 2 | 15 ± 1 | – | – | – |
| 20 | – | – | – | 95 ± 14 | 21 ± 4 | 4 ± 1 | 53 ± 4 | 44 ± 11 | 10 ± 3 |
| 21 | – | – | – | 21 ± 5 | 14 ± 4 | 12 ± 5 | 64 ± 4 | 31 ± 4 | 8 ± 1 |
| 22 | 9 ± 3 | 14 ± 4 | 26 ± 1 | 11 ± 1 | 14 ± 2 | 23 ± 7 | – | – | – |
| borussertib | 2 ± 1 | 37 ± 5 | 394 ± 58 | 6 ± 1 | 15 ± 2 | 46 ± 12 | 91 ± 17 | 30 ± 7 | 6 ± 3 |

Data presented as mean values ± s.d; $n = 3$, where $n$ represents the number of independent experiments.

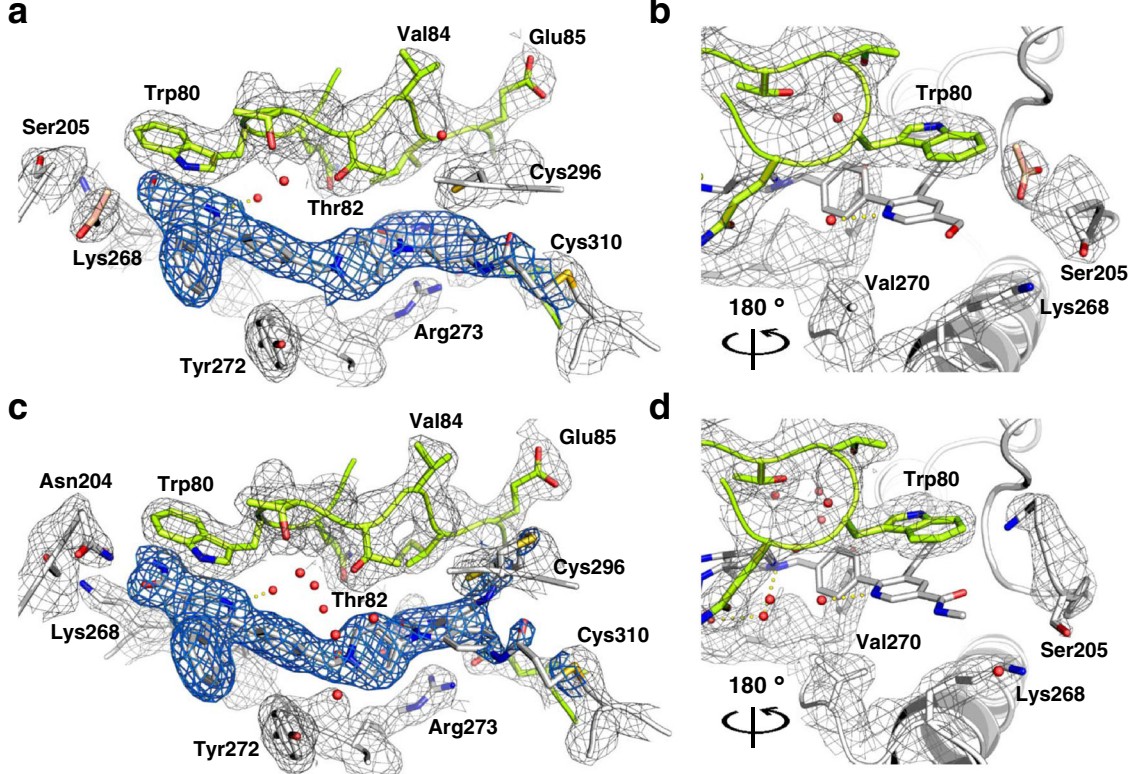

**Fig. 3 Co-crystal structures of full-length Akt1 in complex with pyridine-based Akt1-selective CAAIs, $2F_O-F_C$ maps contoured at 1.0σ. a, b** Co-crystal structure of Akt1 with **3** (PDB: 7NH4). The electron density indicates possible covalent bond formation with both Cys296 and Cys310 but suggests preferred modification of Cys310 (see $F_O-F_C$ simulated annealing omit map in Supplementary Fig. 13).; **c, d** Co-crystal structure of Akt1 with **6** (PDB: 7NH5). The electron density suggested two alternative compound conformations presumably forming a covalent bond with either Cys296 or Cys310 (see $F_O-F_C$ simulated annealing omit maps in Supplementary Fig. 12).

A detailed kinetic analysis of selected CAAIs underlined the strong reversible binding affinities and revealed the contribution of the covalent inactivation rates. Protein mass spectrometry gave orthogonal evidence for the selective irreversible modification. Further structural proof was provided through co-crystallization of Akt1 with potent CAAIs **3** and **6**, which revealed covalent bond formation to the addressed cysteine residues.

We used the established Ba/F3 myr-Akt isoform-dependent system for the cellular evaluation of 23 CAAIs. However, it should be noted that the myristoylation might influence Akt's equilibrium by stabilizing it in its active and open PH-out conformation. Due to the binding mechanism of the presented CAAIs, the conformational changes could reduce binding affinities for the specific myr-Akt constructs as compared to the

wild-type form. Nevertheless, we were able to show a translation of the unique structure-activity relationship of herein-reported inhibitor set into a more complex cellular system by comparing the CTG EC$_{50}$ selectivity log ratios Akt1/Akt2 and Akt2/Akt3 with the corresponding HTRF IC$_{50}$ log ratios. This strategy allows us to select favorable molecules from large inhibitor libraries in a high throughput manner for further, more complex studies. Molecules **1**, **14**, and **20** proved to be particularly promising due to their selectivity for one or two of the isoforms. The results nicely substantiate the previously identified preferences of the isoforms towards different moieties.

In addition, we utilized the human pancreas-derived cancer cell lineage PANC1 as a proof-of-concept for the transferability of the Akt isoform selectivities to a more native cellular system. We

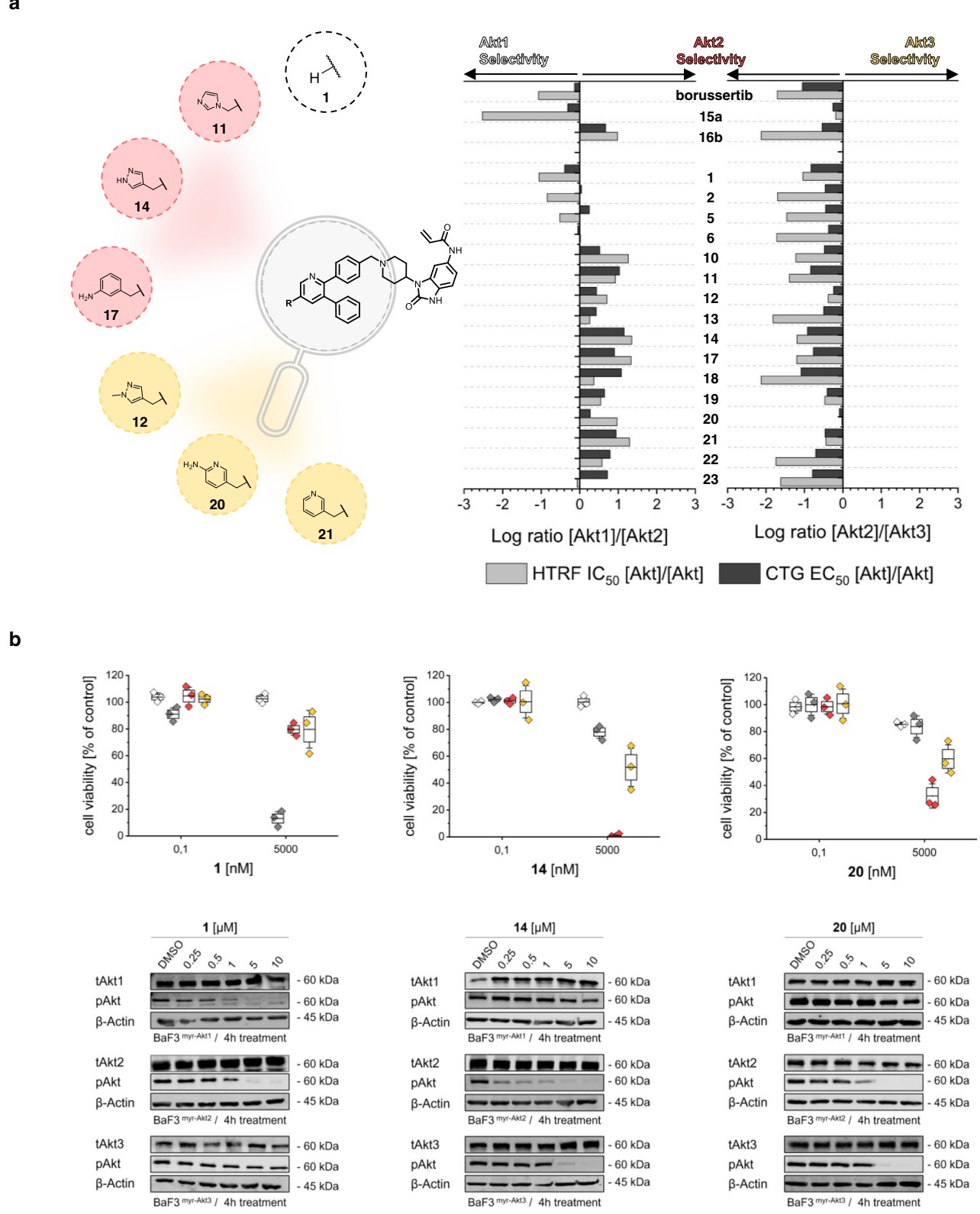

**Fig. 4 Investigation of the Ba/F3 myr-Akt isoform system with a focused set of CAAIs. a** Identified biochemical selectivity profiles (dark gray) compared to the ratios obtained from Cell Viability Assay with Ba/F3 myr-Akt isoform-dependent cells (light gray). Ratios of selectivity in log units are consistent and underline the successful translation of results from biochemical assays to the cellular system. Structures and residues that highlight the observed translation and qualify as potential probe molecules for further investigations. **b** More detailed analysis of inhibitor **1**, **14**, and **20** with the three isoform-dependent cell lines. Graph showing the remaining viable cells with respect to the used concentration of CAAI. Experimental points were measured in duplicates for each plate and were replicated in $n = 3$ biologically independent experiments. The box plot middle line represents the mean value, the box represents the first and third quartile and the error bars indicate s.e. Immunoblots to elucidate dose-dependent downregulation of Akt isoform activity after 4-h treatment ($n = 1$ biologically independent experiments. Source Data are provided with this paper.

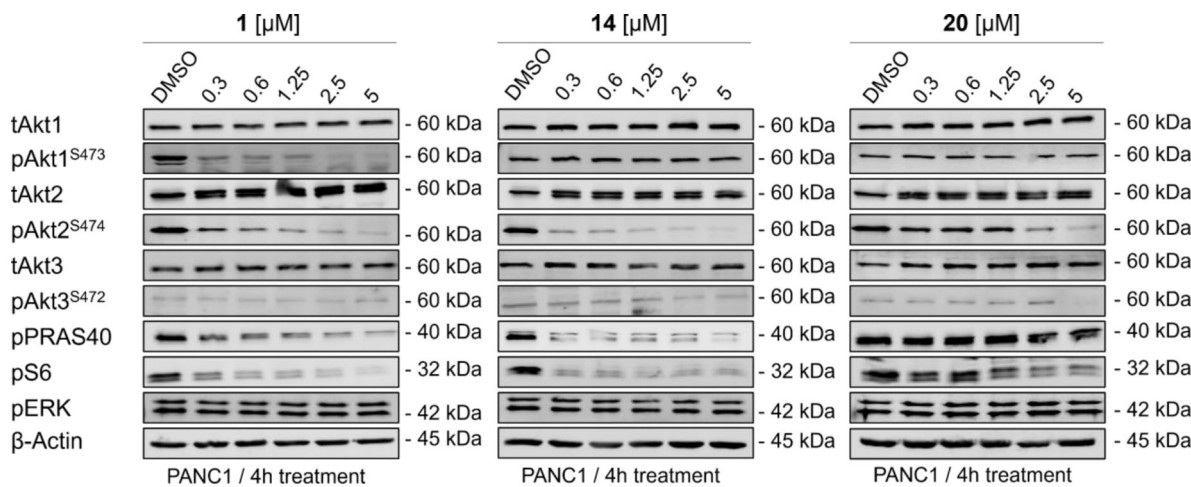

**Fig. 5 Promising Akt isoform-selective inhibitors tested in PANC1 cell line. a** Immunoblots to elucidate dose-dependent downregulation of individual Akt isoform activity after 24-h treatment, with pAkt1 and pAkt2 specific antibodies ($n = 1$ biologically independent experiment). **b** CETSA in PANC1 with inhibitors **1**, **14**, and **20** at different inhibitor concentrations and DMSO as control. The experiment was performed once. Source Data are provided with this paper.

succeeded in proving the selectivity of the molecules **14** for Akt2 and **20** for Akt2 and Akt3 in PANC1 using immunoblot experiments and a cellular thermal shift assay, while molecule **1** was not able to maintain its selectivity for Akt1 in this cell line,

while using high inhibitor concentrations. It is important to note that Akt2 is overexpressed in PANC1 contrary to Akt1 and Akt3, which might have an impact on binding preferences. More in-depth studies in various human cell lines with different Akt

isoform expression profiles are necessary in the future to resolve this issue. These investigations will provide further information regarding the effects of isoform-selective CAAIs on Akt isoforms and their downstream targets.

In summary, we established a straightforward Ba/F3 cellular system to evaluate Akt isoform-selectivity, which translates identified selectivity profiles into complex systems. This simplified model combined with a high throughput readout, serves as a reliable analysis step to identify promising selective inhibitors for subsequent translation into more native cancer model systems. We used the concept of covalent-allosteric inhibitors and utilized those tools to gain unique insights into the structure-activity relationship of the three Akt isoforms. Activity-based characterization and kinetic evaluation revealed a consistent SAR of the covalent-allosteric inhibitors towards the individual Akt isoform inter-domain pockets. This class of inhibitors and their behaviors will guide the development of chemical probes, which can be used as powerful tools within perturbation studies to dissect the isoforms' functions in the context of health and disease, setting ground for functional molecules that can unravel therapeutic advantages[37].

## Methods

**Reagents and materials**. All supplies for the Akt HTRF assay kit were purchased from CisBio (Bagnols-sur-Cèze, France). Active Enzymes were purchased from ProQinase (Akt1 (#1379-0000-2), Akt3 (#1578-0000-1)), and Thermo Fisher Scientific (Akt2 (#PV3184)). Small volume (25 µL fill volume) white round-bottom 384-well plates were obtained from Greiner Bio-One GmbH (Solingen, Germany).

**Sequence alignment and homology modeling**. The sequence alignment of the three Akt isoforms was performed with Clustal Omega[38] based on the following sequence Input files: Akt1_P31749, Akt2_P31751, Akt3_Q9Y243. Homology models for Akt2 and Akt3 were generated with the SWISS-MODEL homology-modelling server based on a Akt1 full-length co-crystal structure (PDB: 6S9X) and the specific sequence files (as mentioned above)[24].

The reported Ligand **16c** was modeled into Akt1 (PDB: 6S9X) with Ligandscout (Inte:Ligand GmbH, Vienna, Austria)[17].

**Activity-based assay**. The biochemical half maximal inhibitory concentrations (IC$_{50}$) were determined with the STK HTRF KinEASE assay (Cisbio) according to the manufactor's instructions. Briefly: 5 µL Kinase solution and 2.5 µL inhibitor solution (8% DMSO in HTRF buffer) were incubated for 1 h before the reaction was started by addition of 2.5 µL starting solution containing ATP and substrate peptide. ATP concentrations were set at their respective $K_M$ values (80 µM for Akt1, 30 µM for Akt2, and 110 µM for Akt3). The following substrate concentrations were used: 250 µM for Akt1 as well for Akt2, and 1 µM for Akt3. After reaction completion (Akt1: 60 min, Akt2: 20 min, Akt3: 15 min), 10 µL of stop solution were added. The FRET signal was measured with an EnVision plate reader (PerkinElmer, Waltham, MA, US) ($\lambda_{ex}$ 620 nm/$\lambda_{em}$ 665 nm). The quotient of both intensities were recorded at eight different inhibitor concentrations and data fit to a Hill 4-parameter equation with Quattro software suite (Quattro Research GmbH, Martinsried, Germany). Each reaction was performed in duplicates and at least three independent determinations of each IC$_{50}$ were made.

**Kinetic Enzyme Assay**. The kinetic evaluation of covalent inhibitors was performed by using PhosphoSens® continuous fluorometric kinase assay (AssayQuant, Marlboro, MA, USA) with the Sox-labeled peptide AQT0535. Phosphopeptide formation was monitored in 20 µL reactions in 384-well plates with a Tecan Infinite M1000 Pro plate reader ($\lambda_{ex}$ 360 nm/$\lambda_{em}$ 485 nm) every 45 s for 1.5 h at room temperature. Reactions were comprised of 10 mM MgCl$_2$, 0.01% Brij-35, 1.25% glycerol, and 1 mM DTT in 54 mM HEPES pH 7.5. ATP concentrations were set at their respective $K_M$ values (140 µM for Akt1, 500 µM for Akt2, and 80 µM for Akt3). By addition of either 8 nM Akt1, 2 nM Akt2, or 5 nM Akt3 (final concentrations) the reactions were initiated. Progression curves were analyzed according to literature procedure using OriginPro (Version 9.7.5.184, OriginLab Corp., MA, USA)[39]. Background substracted progress curves were fit to the following exponential Eq. (1):

$$F = \left( \frac{A}{k_{obs}} \right) \times \left( 1 - e^{-k_{obs} \times t} \right) \qquad (1)$$

where $F$ is the fluorescence, $A$ the amplitude, $k_{obs}$ the rate constant, and $t$ is time. Afterwards the $k_{obs}$-values were plotted versus the inhibitor concentrations $[I]$ and

then fit to the following hyperbolic Eq. (2) to yield $k_{inact}$ and $K_i$:

$$k_{obs} = k_{inact} \times \frac{[I]}{\left( [I] + K_i \left( 1 + \frac{[S]}{K_M} \right) \right)} \qquad (2)$$

where $[S]$ the substrate concentration, and $K_M$ is the substrate concentration that results in half-maximal velocity for the enzyme reaction.

**Protein expression, purification, and crystallization**. A gene encoding for AKT1(2–446, E114/115/116A), AKT2(2–447), and AKT3(2–443, E114/115/116A) including an N-terminal His$_6$-tag and a TEV protease recognition site was synthesized by GeneArt AG and cloned into the pIEx/Bac3 expression vector (Merck Millipore) using *NcoI* and *BamHI* restriction sites. Transfection, virus generation, amplification, and protein expression were carried out in *Spodoptera frugiperda* (Sf9) cells (Thermo Fisher Scientific) following the BacMagic protocol (Merck Millipore). Infected insect cells were grown in Erlenmeyer flasks for 72 h at 27 °C with shaking at 120 rpm. The cells were harvested by centrifugation at 3000×*g* for 20 min and washed once with PBS before being flash frozen in liquid nitrogen. Afterwards, cells were thawed and resuspended in lysis buffer [50 mMol/L Tris, 500 mMol/L NaCl, 1 mMol/L DTT, 0.1% Triton X-100, 10% glycerol, pH 8.0, and EDTA-free protease inhibitor cocktail (Sigma-Aldrich)]. Cells were lysed using a microfluidizer, followed by centrifugation (40,000×*g*, 1 h). The supernatant was loaded on a Ni-NTA Superflow Cartridge (Qiagen). Bound protein was eluted in buffer containing 50 mMol/L Tris, 500 mMol/L NaCl, 500 mMol/L imidazole, 1 mMol/L DTT, 10% glycerol, pH 8.0. For cleavage of the His$_6$-tag, TEV protease was added to the pooled elution fractions and dialyzed overnight into buffer containing 25 mmol/L Tris, 50 mmol/L NaCl, 1 mmol/L DTT, 5% glycerol, pH 8.0 at 4 °C. The cleaved protein was further purified by anion exchange chromatography using a HiTrap Q HP Column (GE Healthcare) followed by size-exclusion chromatography on a HiLoad 16/60 Superdex 75 pg Column (GE Healthcare) using buffer containing 50 mmol/L HEPES, 200 mmol/L NaCl, 1 mMol/L DTT, 10% glycerol, pH 7.3. Afterwards, the protein was transferred into storage buffer (25 mMol/L Tris, 100 mMol/L NaCl, 5 mMol/L DTT, 10% glycerol, pH 7.5) using a Superdex 75 10/300 GL Column (GE Healthcare), concentrated and stored at 80 °C. For crystallization, purified protein at a concentration of 3 mg/mL was incubated with three equivalents of inhibitor on ice for 60 min. The samples were centrifuged at 20,000×*g* for 10 min before hanging drops were prepared in 15-well crystallization plates (EasyXtal Tool, Qiagen) by mixing protein–ligand complex with reservoir solution (1:1) containing 1.25 mMol/L sodium acetate pH 6, 3.75 mMol/L sodium citrate pH 6.5, and 12% PEG MME 2000 at 20 °C. Diffraction-grade crystals grew within 3 days and were cryoprotected using 20% ethylene glycol before they were flash cooled in liquid nitrogen. X-ray diffraction data were collected at the PXII-X10SA beam line of the Swiss Light Source (Paul Scherrer Institute, Villigen, Switzerland) with wavelengths close to 1 Å. The diffraction data were integrated with XDS[40] (X-ray Detector Software) program package and scaled using the program XSCALE[41]. The crystal structure was solved by molecular replacement with PHASER using a co-crystal structure of Akt1 in complex with another CAAI as template[36]. The manual rebuilding of the molecule of the asymmetric unit was performed using the program COOT[42] and with the help of Dundee PRODRG[43] server the inhibitor topology files were generated. For multiple cycles of refinement, phenix.refine[44] was employed and the final structure was evaluated by Ramachandran plot analysis using the server MolProbity[45,46]. Final validation of the model was performed with the help of the PDB_REDO server[47].

**Mass Spectrometry**. We used Akt1, Akt2, and Akt3 for MS experiments and incubated 10 µM of the protein with 100 µM of the inhibitor in a buffer for 1 h and in case of Akt2 2 h. We analyzed the samples by mass spectrometry using a Thermo Fisher Scientific Ultimate 3000 HPLC system connected with a Thermo Fisher Scientific Velos Pro (2D ion trap). The sample (2 µL) was injected and separated by an AdvanceBio Desalting-RP Cartridge (Agilent Technologies) starting at 95 % solvent A (0.1% formic acid in water) and 5% solvent B (0.1% formic acid in acetonitrile) for 30 s, followed by a linear gradient over 2.5 min up to 80% solvent B. A mass range of 700–2000 m/z was scanned and raw data were deconvoluted and analyzed with MagTran (v1.02) or ProMass for Xcalibur (v2.8 rev.2 Novatia).

ESI-MS/MS samples were prepared as stated above followed by steps of reduction, alkylation, and tryptic digest[48]. The digested samples were stage tip-purified using C18 tips and according to a protocol described elsewhere[49]. Subsequently, samples were thawed, dissolved in 20 µL of 0.1% TFA in water, sonicated at room temperature for 15 min, and centrifuged at 15,000×*g* for 1 min shortly before analysis. In all, 3 µL of sample were loaded onto a pre-column cartridge and desalted for 5 min using 0.1% TFA in water as eluent at a flow rate of 30 µL/min. The samples were back-flushed from the pre-column to the nano-HPLC column during the whole analysis. Elution was performed using a gradient starting at 5% B with a final composition of 30% B after 35 min (flow rate 300 nL/min) using 0.1% formic acid in water as eluent A and 0.1% formic acid in acetonitrile as eluent B and a column temperature of 40 °C. The nano-HPLC column was washed by increasing the percentage of solvent B to 60% in 5 min and to 95% in additional 5 min, washing the columns for further 5 min, flushing back to starting conditions and equilibration of the system for 14 min. During the complete gradient cycle, a typical TOP10 shot-gun proteomics method for the MS and MS/

MS analysis was used. For full scan MS experiments a mass range of *m/z* 300–1650 was scanned with a resolution of 70000. MS/MS experiments were followed by up to 10 high energy collision dissociation (HCD) MS/MS scans with a resolution of 17,500 of the most intense at least doubly charged ions. Data evaluation was performed using MaxQuant[50]. Spectra were searched against the specific Akt sequence and a contamination database using a false discovery rate of 1% on peptide and protein level using a decoy database for determination of the false discovery rate. For database search oxidation of methionine and N-terminal acetylation of proteins, carbamidomethylation of cysteines, and artificial modification of cysteines were defined as variable modifications.

**Myr-Akt expression constructs**. 1051 pBABEpuroL-AKT1, 1271 pBABEpuroL-Myr-HA-AKT2, and 1272 pBABEpuroL-Myr-HA-AKT3 were a gift from William Sellers (Addgene plasmid #9011/ 9018/ 9019; http://n2t.net/addgene:90011/ 9018/ 9019; RRID: Addgene_ 9011/ 9018/ 9019). Wild type version of human AKT1 coding sequence derived from pBABEpuroL-AKT1 was cloned by PCR. The DNA inserts were cloned into the BamHI and EcoRI sites of the pBABEpuroL-Myr-HA retroviral transfer vector (derived from pBABEpuroL-Myr-HA-AKT2, by excising the AKT2 coding sequence). pCLEco retrivirus packaging vector was a gift from Martin Sos (University of Cologne). Primer sequences are listed in the supplementary information table 4. All sequences were verified by sequencing both strands. Constructs were packaged into VSV-G envelope pseudotyped retroviruses by transiently transfecting HEK293T cells using TransIT-LT (Mirus). Cells were selected and maintained in puromycin (5 ng/µL).

**Cell lines**. HEK293T cells were a gift from Dr. Daniel Summerer (Technical University Dortmund). Ba/F3 cells were obtained from Dr. Martin Sos (University of Cologne). Cells were maintained in DMEM or RPMI-1640 medium (Gibco) supplemented with 10 % fetale bovine serum (FBS) (PAN-Biotech) and 1% penicillin-streptomycin (Gibco). The parental Ba/F3 cells were supplemented with 10 ng/mL IL-3 (PeproTech). All cells were cultured at 37 °C at 5% $CO_2$.

**Generation of the retrovirus**. To infect Ba/F3 cells, retrovirus was generated by liposomal transfection into HEK293T cells. For this purpose, $7.5 \times 10^4$ HEK293T cells were first seeded in a dish in 5 mL of cultivation medium (RPMI-1640 (Gibco), 10% FBS (PAN-Biotech), 1% penicillin/streptomycin (Gibco)). After 24 h, the transfection mixture (250 µL OptiMEM (Gibco), 2.5 µg pBABE construct, 2.5 µg pCLEco, and 7.5 µL TransIT-LT (Mirus)) was added to the HEK293T cells. After 72 h of incubation at 37 °C and 5% $CO_2$ the virus was harvested by centrifugation of the cell suspension at 180 g. The virus-containing supernatant was sterile filtered using a 0.45 µm syringe filter and used directly for infection of Ba/F3 cells or stored at −80 °C.

**Genome isolation and gene sequencing**. To verify the correctly infected Ba/F3 cells, genome isolation and subsequent sequencing focusing on the target gene was performed. First, the genomic DNA of the Ba/F3 cells was isolated and subsequently the target gene sought was specifically amplified by PCR using target gene flanking primers.

**CRISPR/Cas9-mediated knockout of AKT1/2**. PANC1 cells were first stably transduced with lentiviral particles produced in HEK293T cells using the following vectors: psPAX2 (Addgene, #12260) and pCMV-VSV-G (Addgene,#8454), lentiCas9-Blast (Addgene, #52962) and Lipofectamine LTX and Plus Reagent (Life Technologies). Cells were selected by growth in the presence of 7.5 µg/ml blasticidin and surviving colonies were analyzed for successful CAS9 expression by western blot. PANC1-Cas9 cells were then seeded in six-well plates ($2.5 \times 10^5$) to reach a confluency of 70–80% at the day of transduction. For performing AKT1 and AKT2 knockdown, 1.5 µg of respectively two sgRNAs targeting AKT1 (Addgene, #75500) and AKT2 (Addgene, #77505) were used for producing lentiviral particle (description before). Seeded PANC1-Cas9 cells were transduced with those lentiviral particles and selected by growth in presence of 1 µg/ml of puromycin for 14 days. Surviving colonies were analyzed for successful knockdown by western blot and qRT-PCR (Supplementary Fig. S14). For RNA isolation the Maxwell RSC simplyRNA Cells Kit was used. Complementary DNA (cDNA) synthesis was performed by following the TaKaRa PrimeScript™RT Master Mix reverse transcription reaction protocol. Primer sequences were provided by Prof. Dr. Stephan Hahns working group in Bochum (AKT1) and from literature (AKT2)[51]. For calculation of the relative gene expression the delta-delta CT method was used. β-Glucuronidase (GUSB) was used as a reference gene.

**Cell Viability Assay**. On day 0, cells were plated into white 384-well cell culture plates (Greiner Bio-One) using a Multidrop reagent dispenser (Thermo Fisher Scientific) at cell numbers that ensure linear and optimal luminescent signal intensity (Ba/F3myr-Akt1: 600 cells/well, Ba/F3myr-Akt2: 800 cells/well, Ba/F3myr-Akt3: 800 cells/well). Following incubation for 24 h in a humidified atmosphere at 37 °C/5% $CO_2$, cells were treated with inhibitors in serial dilutions ranging from 30 mMol/L down to 0.1 nMol/L using an Echo 520 acoustic liquid handler (Labcyte Inc.). Cell viability was analyzed on day 4 using the CellTiter-Glo

Assay (Promega) as per the manufacturer's instructions. Luminescence was recorded using an EnVision Multilabel 2104 Plate Reader (PerkinElmer) using 500 ms integration time. The obtained data were normalized to the plate positive control (30 mMol/L staurosporine) and negative control (DMSO) and subsequently analyzed and fitted with the Quattro Software Suite (Quattro Research) using a four-parameter logistic model. All experimental points were measured in duplicates for each plate and were replicated in at least three plates.

**Western blot analysis**. Cells were seeded into six-well tissue culture plates (Sarstedt) (BaF3 myr-Akt1/2/3: $10^6$ cells/well, PANC1: 250.000 cells/well). After 24 h incubation, cells were treated with various concentrations of inhibitors or DMSO and incubated for additional 4.5 h before cells were washed twice with ice-cold PBS. Cell lysis was initiated by addition of 100 µL RIPA buffer (Cell Signaling Technology) per well supplemented with phosphatase and protease inhibitor cocktails (Sigma) followed by incubation on ice for 20 min. Cells were then harvested by scraping (PANC1) or centrifugation (BaF3). Whole cell lysates were cleared by centrifugation at 14,000×*g*/4 °C for 10 min and transferred into fresh, pre-cooled microcentrifuge tubes. Protein concentrations were determined using the Pierce BCA protein assay (Thermo) as per manufacturer's instructions. Equal amounts of protein were separated by SDS-PAGE and transferred to Immobilon-FL PVDF membranes (Merck Millipore) using Pierce™ 1-step transfer buffer (Thermo) and the Pierce™ Power Blotter (Thermo). Membranes were washed for 5 min with ddH2O, blocked with Intercept®Blocking Buffer TBS (Li-Cor) for 1 h at room temperature, and then incubated with primary antibodies diluted in Intercept®Blocking Buffer TBS overnight at 4 °C with gentle agitation. On the next day, the membranes were washed three times with TBS-T (50 mM Tris, 150 mM NaCl, 0.05 % Tween 20, pH 7.4) for 5 min before being incubated with secondary antibodies diluted in Intercept®Blocking Buffer TBS for 1 h at room temperature with gentle agitation. Finally, the membranes were washed three times for 5 min with TBS-T and then scanned using an Odyssey®CLx imaging system (Li-Cor).

**Cellular thermal shift assay**. Cells were seeded into 100 mm tissue culture plates (Sarstedt) (PANC1: $10^6$ cells). After 24 h incubation, cells were treated with 1 µM of inhibitors or DMSO and were incubated for 4.5 h. Cells were washed twice with ice-cold PBS and then suspended in 1 mL PBS. Cells were collected and separated into equal volumes (100 µL). The aliquots were treated individually with different temperatures for 3 min, followed by a 3 min incubation at room temperature. Cell lysis was initiated by adding 100 µL RIPA buffer and 2 freeze-thaw cycles in liquid nitrogen. The lysates were centrifuged at 21,130×*g* for 20 min at room temperature. The supernatants were transferred into fresh, pre-cooled microcentrifuge tubes, and analyzed by western blot analysis as described above.

**Antibodies**. Anti-pAkt(Ser473) (CST, cat. no. 4060, 1:1000), anti-tAkt1 (CST, cat. no. 2938, 1:1000), anti-tAkt2 (CST, cat. no. 3063, 1:1000), anti-tAkt3 (CST, cat. No. 3788, 1:1000), anti-pAkt1 (CST, cat. No. 9018, 1:1000), anti-pAkt2 (CST, cat. No. 8599, 1:1000), anti-pAkt3 (Thermo Fisher Scientific, cat. No. PA5-12898, 1:1000), anti-pPRAS40(Thr246) (CST, cat. no. 2997, 1:1000), anti-pS6(Ser235/236) (CST, cat. no. 2317, 1:2000), anti-pERK1/2(Thr202/204) (CST, cat. No. 4370, 1:1000), anti-pFOXO (CST, cat. no. 2599, 1:500), anti-pGSK3β (CST, cat. no. 5558, 1:1000), anti-cPARP/PARP (CST, cat. no. 9542, 1:1000), anti-β-Actin (Sigma, cat. no. A5441, 1:5000), and anti-mouse IgG (H+L) (DyLight™ 680 Conjugate) (CST, cat. no. 5470, 1:15,000), anti-rabbit IgG (H+L) (DyLight™ 800 4X PEG Conjugate) (CST, cat. no. 5151, 1:15,000).

**Reporting summary**. Further information on research design is available in the Nature Research Reporting Summary linked to this article.

## Data availability
The data supporting the findings of this study are available in the paper and its Supplementary Information. Source Data are provided with this paper. The crystal structure data generated in this study have been deposited in the PDB database under accession code 7NH4 and 7NH5. Original diffraction data can be accessed on https://www.proteindiffraction.org/ under the same accession codes. Already reported structures are deposited under following accession codes: 1MRY, 2UVM, 4GV1, 6HHF, and 6S9X. Source data are provided with this paper.

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

## Acknowledgements

L.Q. is grateful for a scholarship by the German Academic Scholarship Foundation. Dr. Petra Janning, Andreas Brockmeyer, and Malte Metz are thanked for the support during the MS/MS measurements. We thank Dr. Martin Sos (University of Cologne) for the Ba/F3 cells. This work was co-funded by the German Federal Ministry for Education and Research (e:Med, Grant No. BMBF 01ZX1901B), the German federal state North Rhine-Westphalia (NRW), and the European Union (European Regional Development Fund: Investing In Your Future) (EFRE-0200479, EFRE -0400199) Drug Discovery Hub Dortmund (DDHD) and KomIT. J.T.S. is supported by the German Cancer Consortium (DKTK), the Deutsche Forschungsgemeinschaft (DFG) grant SI1549/3-1 (DFG/GRC-CRU337) and SI1549/4-1, and the German Cancer Aid (grant no. 70112505; PIPAC consortium). F.G. is supported by the German Cancer Aid (#70113128).

## Author contributions

D.R. is responsible for initiating and supervising the project. L.Q. designed and together with T.K., J.N., L.M.L., and M.K. synthesized the compounds. L.Q. carried out the biochemical experiments. L.D., I.L., and M.L. performed the cell biology studies. I.L. and L.D. carried out the structural biology studies. F.G. and K.A. generated the PANC1

knockout cell lines. N.U., J.W., C.S.F., M.P.M., and J.T.S. advised the study. The manuscript was written through contributions of all authors. All authors have given approval to the final version of the manuscript.

## Funding

## Competing interests
The authors declare no competing interests.
