## [Peer Review File · Nature Communications]

REVIEWER COMMENTS

Reviewer #1 (Remarks to the Author):

In this study, Dr. Daniel Rauh and colleagues have developed cellular expression systems for three AKT isoforms (AKT1/2/3) and then designed and tested a series of AKT covalent inhibitors based upon their previously reported AKT inhibitor and showed that some of these compounds can achieve moderate isoform selectivity. They also determined co-crystal structures of AKT1 in complex with two designed inhibitors (compounds 1 and 3). Furthermore, the authors tested 3 AKT inhibitors with different isoform selectivity profiles in the PANC1 cancer cell line and showed that the selectivity obtained from the model systems can be validated in the PANC1 cell line.

This is a quite comprehensive study and the conclusions are supported by the data. The authors have developed useful cellular systems to evaluate the AKT activity for 3 different AKT isoforms and have validated the cellular systems through the design and evaluation of a new set of covalent AKT inhibitors. Overall, while the overall innovation is moderate, this is a useful contribution to the AKT field and to the large scientific community. It is recommended that the paper be published after the following revisions.

1. Since the authors have identified reasonably selective AKT1, AKT2 and AKT3 inhibitors, it will be nice to demonstrate the potential therapeutic utilities of such inhibitors;
2. In Figure 5, the authors have shown that compounds 14 and 20 can stabilize AKT2 and AKT3 in cells by CETSA. It will be useful to show that selective AKT1 inhibitor can stabilize AKT1 in cells and pan AKT1/2/3 inhibitors can stabilize AKT1/2/3 in cells.
3. Through medicinal chemistry efforts, the authors have identified AKT inhibitors with moderate selectivity. It will be much more impactful to discover AKT isoform selective inhibitors with at least >100-fold selectivity.

Reviewer #2 (Remarks to the Author):

In this manuscript, Lena Quambusch et al. attempt to develop a stable Akt isoform dependent cell model to characterize a novel set of Akt isoform-selective covalent allosteric inhibitors as a continuation of previously published work (Quambusch et al. *Angew Chem* 2019). The authors employed the murine pro-B Ba/F3 cell line which is highly dependent on the exogenous supply of IL-3, and used retrovirus system to have transiently overexpressing of each of constitutively active Akt isoforms 1-3 (Akt_s) with N-ter myristoylation. Next, they characterized the Ba/F3 cell sensitivity to their previously published Akt covalent inhibitor borussertib and ATP-competitive inhibitor capivasertib to prove that the cells highly depends on transiently expressed myr-Akt isoform and independent of IL-3 induced signaling, despite the endogenous Akt_s still present in the cells. Moreover, they also performed a detailed study of structure-activity relationship with a focus on obtaining Akt isoform selective covalent molecules. A combination of X-ray crystallography, biochemical and kinetic assays revealed a novel interesting set of Akt isoform selective inhibitors such as compound #1 for Akt1, compounds # 11, 14, 22 for Akt2, compound # 20 for Akt_s 2, 3. Perhaps, this paper provides very useful tools for proteomic studies to help understanding the function of each Akt isoforms 1 and 2 that have overlaps and are unclear in cancer progressions.

Overall, this is a an interesting paper with puzzling results that require further exploration to make solid conclusions. It requires at the very least to address the following

1. For Fig. 1b, the authors wanted to prove that the Ba/F3 cell model only responds to myr-Akt isoform via testifying well-established Akt substrate PRAS40 pT246, but the parental cell line also showed equal level of PRAS40 pT246. This data showed an inconsistency to the interpretation in lines 77-79. The author should clarify this confusion and also analyze with other well-known substrates such as FOXO1 or 3A. How do they distinguish the Ba/F3 PI3K/Akt signaling only from transfected myr-Akt isoform but not from endogenous Akt_s? How does myr-Akt isoform stucked to the plasma membrane phosphorylate its nucleus PRAS40 substrate?

Proof of cell identification will be required

2. For Fig 5a, in order to prove the promising of the Akt isoform selective inhibitors, they performed cell-based assays with pancreatic cancer cell line PANC1 with 3 compounds 1, 14 and 20 and showed that blocking the activation of selected Akt isoform for instance, Akt2 is good enough to block the Akt downstream signaling. In this context, they showed that the compound 14 only blocked Akt2 pSer474 but not Akt1 pS473 significantly reduced PRAS40 pT246. Since PRAS40 Thr246 is a long well-established Akt1 substrate in both in vivo and in vitro assays, can the authors clarify that the active pSer473 Akt1 in the PANC1 cells does not phosphorylate PRAS40 Thr246?

For both points 1 and 2, the authors should use Akt1/2 knock out HCT116 cell line (PMID: 20133737) and test other Akt substrates to confirm their cell-based results. The results present with the new cell line should be compared to the known Akt1/2 knock out

3. For the kinetics of Akt covalent inhibitors, the author should provide the protocol and data in supplementary to support the Table 2.

The authors carried out homology models for akt2 and akt3 based on the akt1 full length structure. They use the models to hypothesize that changes on the chemical nature and properties of the residue at position 169 induce changes in the electrostatic potential in the different isoforms.

4. Most of the methods are not described enough. They should be modified with the required detail that would allow reproducibility. Examples: activity based assay, cell viability assay. Furthermore, the structure determination should be described with details and references of the software packages used.

5. The captions should be corrected and all yellow highlighting should be removed (caption figure 3); Co-crystal structure of Akt1 with 6 (PDB: 7NH5). The sentence "The electron density indicated two alternative compound conformations" should be modify to 'suggested'

Minor comments and typos

The title of Akt1 co-crystal structures confirm binding mode is not a good title.

Line 261 needs a 'to' in ...selectivities to a more native..."

265; not a good statement from the authors : For this reason, more in-depth studies in various human

266 cell lines with different Akt isoform expression profiles are necessary and ongoing.

In methods reorganize to have reagents and materials before all methods

The activity based assay should be described.

A space between the numeral and the units should be added (for example line 314 50mMol/L

The font with outline and color inside used to label figures (sup fig s2) should be changed, is difficult to read.

Figure 3, all 4 panels should have letters to ease description

Sup table 1 (crystallography) should include ‘,’ in large numbers such as 20,959 and the same number of decimals throughout

Reviewer #3 (Remarks to the Author):

The protein kinase Akt plays a critical role in the PI3K/Akt/mTOR signaling pathway and is therefore an important drug target. While traditional approaches such as genetic deletions are limited in discerning the different roles each of the three Akt isoforms plays, authors of this manuscript propose a novel cellular model system that allows the evaluation of the isoform selectivity profile of Akt inhibitors in a cellular context, complementary to biochemical assays. This work provides a tool to assess and aid the design of isoform-specific Akt inhibitors, and therefore is potentially of high impact and great interest to a broad group of scientists.

The manuscript is in general well written and easy to follow. I believe the manuscript is worthy of being published in Nature Communications after a revision with the following points being addressed:

Major points:

1. In the discussion of Figure 1b, the authors claim that the level of pPRAS40 indicates the activation of downstream signaling pathways by myr-Akt. However, in the parental system the pPRAS40 level is high despite the extremely low level of pAkt, and also in Figure S1b, the pPRAS40 level doesn't seem to correlate with the pAkt level --- do these results suggest alternative signaling pathways (other than the Akt-dependent pathway) in the cellular system that are responsible for the phosphorylation of PRAS40? It's necessary that the authors exclude this possibility; otherwise pPRAS40 won't be a useful indicator of Akt-triggered cell signaling.
2. In the immunoblot of Figure 1b, the myr-Akt1/2/3 cell lines all show significantly lower pS6 level (consistent with slower growth rate), however, in Figure 1c, all three myr-Akt cell lines show significant amount of pS6 without inhibitor treatment (the DMSO panel). How to explain the discrepancy in the baseline pS6 level between Figure 1b and 1c?

3. In Figure 1d: the authors are trying to show the difference in potency of an inhibitor against various Akt isoforms by showing the pAkt level and cell viability upon treatment. Since pErk and pS6 have been shown in previous figures to indicate cell proliferation level, it's necessary to also show pErk and pS6 levels here in Figure 1d (expected to be very low) to confirm the low cell viability.
4. The authors build homology models of Akt2 and 3 (based on an Akt1 crystal structure) and make comments on specific intramolecular interactions, solvent accessibility and the size of the interdomain pocket of the models, which seems insufficiently supported. It's necessary that the authors provide evidence such as short molecular dynamics simulations showing the stability of the homology models, or structural support from existing Akt2/3 crystal structures?
5. Line 171: How to explain the different inhibitor preference of Akt2 and Akt3? If Akt2 favors positively charged functional groups because of Asp269, can we hypothesize that Akt3 favors negatively charged groups due to the Lys in the corresponding position? Can this be proven by inhibitors with negatively charged groups?
6. The authors use molecule 20 as a preferred inhibitor for Akt3, which is not very accurate because the IC50, EC50, cell viability and CETSA stability results all indicate more inhibition by molecule 20 of Akt2 than Akt3.

Minor points:

1. Figures such as Figure 1c and 1d are low resolution and it's difficult to read the labels below the immunoblot panels.
2. There's a mixture of using "mol/L" and "M" (Molar) in Figure 1c/d and throughout the Methods section, it'd be much clearer if the authors unify them.
3. In lines 92-93, the authors mentioned IC50 values of 500 nM, 1nM and 12 nM for Akt3, Akt1 and Akt2, respectively. If these IC50 values are from biochemical assays, the results need to be shown explicitly.
4. When the authors describe kinase structures in the manuscript, it would be very helpful if they'd refer to canonical structural features such as the hinge region, the C helix, the DFG motif, the activation loop etc, instead of only referring to residue indices since the numbering system differs from one kinase to another.
5. The structural representations throughout this manuscript aren't clear enough. For example: Figure 2b seems a zoom-in of part of Figure 2a, however, the view seems rotated (180 deg?), which can be confusing for readers; in Figure 2C, the "enlarged space in Akt2 and Akt3" needs to be explicitly pointed to; also, in some figures (e.g. Figure 2d, S2A) some explicitly shown residues (and mentioned in the text) are not labeled, which makes it hard to locate them.
6. Line 203: "selectivity ratios" should be "selectivity log ratios".

Response to Decision Letter: Quambusch, Depta, et al.

Manuscript NCOMMS-21-12507-T

Reviewer: 1

This is a quite comprehensive study and the conclusions are supported by the data. The authors have developed useful cellular systems to evaluate the AKT activity for 3 different AKT isoforms and have validated the cellular systems through the design and evaluation of a new set of covalent AKT inhibitors. Overall, while the overall innovation is moderate, this is a useful contribution to the AKT field and to the large scientific community. It is recommended that the paper be published after the following revisions.

We are grateful and thank the reviewer for the positive assessment of our work.

1. Since the authors have identified reasonably selective AKT1, AKT2 and AKT3 inhibitors, it will be nice to demonstrate the potential therapeutic utilities of such inhibitors.

We agree with the reviewers' comment regarding a therapeutic utility of our compounds. Anyhow, as the editor kindly stated, such an endeavor would lie beyond the scope of this paper.

2. In Figure 5, the authors have shown that compounds 14 and 20 can stabilize AKT2 and AKT3 in cells by CETSA. It will be useful to show that selective AKT1 inhibitor can stabilize AKT1 in cells and pan AKT1/2/3 inhibitors can stabilize AKT1/2/3 in cells.

The referee brings up an excellent point. We now include additional data from CETSA experiments using the potent covalent-allosteric Akt inhibitor borussertib and lower concentrations of our novel Akt1 selective inhibitor 1 in PANC1 cells (Figure 5b and S16). The new data with inhibitor 1 underlines a more profound Akt1 stabilization at the cell level.

3. Through medicinal chemistry efforts, the authors have identified AKT inhibitors with moderate selectivity. It will be much more impactful to discover AKT isoform selective inhibitors with at least >100-fold selectivity.

This point is well taken, and we are currently focusing on another round of improved CAAs guided by our presented design and results. Yet again, this task demands more time and lies beyond the scope of this study.

Reviewer: 2

Overall, this is an interesting paper with puzzling results that require further exploration to make solid conclusions. It requires at the very least to address the following:

1. For Fig. 1b, the authors wanted to prove that the Ba/F3 cell model only responds to myr-Akt isoform via testifying well-established Akt substrate PRAS40 pT246, but

the parental cell line also showed equal level of PRAS40 pT246. This data showed an inconsistency to the interpretation in lines 77-79. The author should clarify this confusion and also analyze with other well-known substrates such as FOXO1 or 3A. How do they distinguish the Ba/F3 PI3K/Akt signaling only from transfected myr-Akt isoform but not from endogenous Akt?

We thank the referee for his/her comment. We agree that the wording in lines 77-79 is misleading and therefore rephrased it. Additionally, we analyzed the substrates FOXO and GSK3 β (Figure 1b), showing modified phosphorylation levels due to the implementation of myr-Akt isoforms. Overall, it has to be noted that this is an artificial cellular system and derived from a murine origin. For this reason, we believe that any conclusions about downstream effects are limited and should be viewed with caution.

The text now reads as follows: "Additionally, the myr-Akt isoforms trigger an increased phosphorylation of the downstream targets FOXO and GSK3 β , while maintaining the phosphorylation level of PRAS40, indicating the activation of downstream signaling pathways."

How does myr-Akt isoform stucked to the plasma membrane phosphorylate its nucleus PRAS40 substrate?

The reviewer raises important and interesting questions regarding the localization of myristoylated Akt in cells. More in-depth localization and cell biological studies would be required to resolve this issue. However, we feel this is beyond the scope of our manuscript. In this study, we intended to use the lipidation tag to increase the activated fraction of the transfected oncogene in the cells and show the applicability of the engineered cellular model system for the high-throughput evaluation of potential Akt inhibitors.

Proof of cell identification will be required.

The requested sequencing data are now provided with the source data file, clearly showing the presence of myr-Akt constructs.

2. For Fig 5a, in order to prove the promising of the Akt isoform selective inhibitors, they performed cell-based assays with pancreatic cancer cell line PANC1 with 3 compounds 1, 14 and 20 and showed that blocking the activation of selected Akt isoform for instance, Akt2 is good enough to block the Akt downstream signaling. In this context, they showed that the compound 14 only blocked Akt2 pSer474 but not Akt1 pS473 significantly reduced PRAS40 pT246. Since PRAS40 Thr246 is a long well-established Akt1 substrate in both in vivo and in vitro assays, can the authors clarify that the active pSer473 Akt1 in the PANC1 cells does not phosphorylate PRAS40 Thr246?

For both points 1 and 2, the authors should use Akt1/2 knock out HCT116 cell line (PMID: 20133737) and test other Akt substrates to confirm their cell-based results. The results present with the new cell line should be compared to the known Akt1/2 knock out.

We agree with the reviewer that the investigation of knockout models is an important cross-validation for evaluating inhibitor-mediated findings. To address this concern, we evaluated the Akt-dependent signaling in PANC1 cells harboring either an Akt1 or an Akt2 knockout. These cell lines are well

established in our lab (doi: 10.1002/anie.201909857) and enabled the direct evaluation of the observed phenotypes in the already used parental PANC1 cells. The ko models revealed a decreased pPRAS40 level for both - Akt1 ko and Akt2 ko, qualifying it as a suitable marker for the impairment of either one of the two isoforms. In contrast, the pS6 levels are not downregulated in the PANC1 ko cell lines, which hints at possible compensation mechanisms in response to gene knockout, as discussed in further studies (doi: 10.1016/j.tibs.2011.03.006).

3. *For the kinetics of Akt covalent inhibitors, the author should provide the protocol and data in supplementary to support the Table 2.*

We provide the protocol in more detail, added the source data to support table 2, and give examples of resulting plots in the supplementary information, which will help understand the assay principle and data analysis.

4. *Most of the methods are not described enough. They should be modified with the required detail that would allow reproducibility. Examples: activity based assay, cell viability assay. Furthermore, the structure determination should be described with details and references of the software packages used.*

We now include more detailed descriptions of the aforementioned methods and state the software packages used for the structure determination.

5. *The captions should be corrected and all yellow highlighting should be removed (caption figure 3); Co-crystal structure of Akt1 with 6 (PDB: 7NH5). The sentence "The electron density indicated two alternative compound conformations" should be modify to 'suggested'.*

We corrected for this.

Minor comments and typos

The title of Akt1 co-crystal structures confirm binding mode is not a good title.

In this point, we respectfully disagree with the referee. In respect to already published Akt1 co-crystal structures, we think the here presented complexes confirm our inhibitor design approach and anticipated binding mode.

Line 261 needs a 'to' in ...selectivities to a more native..."

We corrected for this.

265-266; not a good statement from the authors: For this reason, more in-depth studies in various human cell lines with different Akt isoform expression profiles are necessary and ongoing.

The sentence now reads: "More in-depth studies in various human cell lines with different Akt isoform expression profiles are necessary in the future to resolve this issue."

In methods reorganize to have reagents and materials before all methods

We rearranged the methods as suggested.

The activity based assay should be described.

A more detailed description is now included.

A space between the numeral and the units should be added (for example line 314 50mMol/L.

This has been corrected.

The font with outline and color inside used to label figures (sup fig s2) should be changed, is difficult to read.

We increased the size and changed the labels.

Figure 3, all 4 panels should have letters to ease description

We corrected for this.

Sup table 1 (crystallography) should include ',' in large numbers such as 20,959 and the same number of decimals throughout.

We changed the punctuation and decimals.

Reviewer 3:

This work provides a tool to assess and aid the design of isoform-specific Akt inhibitors, and therefore is potentially of high impact and great interest to a broad group of scientists. The manuscript is in general well written and easy to follow. I believe the manuscript is worthy of being published in Nature Communications after a revision with the following points being addressed:

We are grateful and thank the reviewer for the enthusiastic assessment of our work.

1. In the discussion of Figure 1b, the authors claim that the level of pPRAS40 indicates the activation of downstream signaling pathways by myr-Akt. However, in the parental system the pPRAS40 level is high despite the extremely low level of pAkt, and also in Figure S1b, the pPRAS40 level doesn't seem to correlate with the pAkt level --- do these results suggest alternative signaling pathways (other than the Akt-dependent pathway) in the cellular system that are responsible for the phosphorylation of PRAS40? It's necessary that the authors exclude this possibility; otherwise pPRAS40 won't be a useful indicator of Akt-triggered cell signaling.

We thank the reviewer for this excellent point and refer to our response to reviewer #2 (1.)

2. In the immunoblot of Figure 1b, the myr-Akt1/2/3 cell lines all show significantly lower pS6 level (consistent with slower growth rate), however, in Figure 1c, all three myr-Akt cell lines show significant amount of pS6 without inhibitor treatment (the DMSO panel). How to explain the discrepancy in the baseline pS6 level between Figure 1b and 1c?

The referee brings up an excellent point. The discrepancy in the pS6 levels can be explained due to different contrast settings used for the pS6 panel in Figure 1b and the pS6 panel in Figure 1c. We chose to increase the contrast for the pS6 panel in figure 1c to better visualize the dose-dependent downregulation of the residual S6 signal. We added an example to the source data file.

3. In Figure 1d: the authors are trying to show the difference in potency of an inhibitor against various Akt isoforms by showing the pAkt level and cell viability upon treatment. Since pErk and pS6 have been shown in previous figures to indicate cell proliferation level, it's necessary to also show pErk and pS6 levels here in Figure 1d (expected to be very low) to confirm the low cell viability.

To address this concern, we included an example western blot in the SI showing reduced pS6 levels due to treatment with inhibitor 15a (Figure S2). Since pErk is part of another signaling pathway and is unaffected by Akt inhibition, we decided to include this marker only in the initial validation.

4. The authors build homology models of Akt2 and 3 (based on an Akt1 crystal structure) and make comments on specific intramolecular interactions, solvent accessibility and the size of the interdomain pocket of the models, which seems insufficiently supported. It's necessary that the authors provide evidence such as short molecular dynamics simulations showing the stability of the homology models, or structural support from existing Akt2/3 crystal structures?

We thank the referee for this comment. To provide further evidence, we now included a comparison of the Akt2 Model with a literature known crystal structure of the inactive kinase domain of Akt2 (pdb: 1mry) in the supplementary information. Further, we diminished the above-mentioned sentence within the main text. It now reads: "The model of Akt2 is in good agreement with a crystal structure of the inactive kinase domain (PDB: 1mry, Supplementary Fig. S3)."

In addition, we would like to mention that these models (successfully) guided our small molecule design to obtain isoform-selective molecules, thus proving that the initial assumptions must (at least to a certain extend) have been correct.

5. Line 171: How to explain the different inhibitor preference of Akt2 and Akt3? If Akt2 favors positively charged functional groups because of Asp269, can we hypothesize that Akt3 favors negatively charged groups due to the Lys in the corresponding position? Can this be proven by inhibitors with negatively charged groups?

We thank the reviewer for looking into the SAR in detail. Indeed, the design of inhibitors inheriting a negatively charged group is planned to elaborate on this hypothesis. Based on our homology models, we think an interaction of Lys266 in Akt3 might be out of reach for our current inhibitor series. The altered amino acid Gly265 allows to bind bulkier residues in a different position, next to

Tyr261 (see SI Fig S4). Further, the SAR shows that some positive charges are tolerated in Akt3 (Molecule 14, 19), which excludes a direct repulsion of Lys266.

Due to the altered helical structure of the α E-helix in Akt2, it seems that this isoform possesses a hybrid-like character between Akt1 and Akt3. At least this assumption would support the equipotency of amide-linked molecules such as 22, binding Akt1 and Akt2. Whereas methyl-linked molecules such as 21 binds only Akt2 and Akt3. To elaborate on the different inhibitor preferences, we now include the following sentence: “Interestingly, Akt2 seems to tolerate both linkages, amide or methylene, whereas Akt1 prefers the amide-linked version and Akt3 the methyl ones, e.g., pyridine 21 and 22. With respect to the model-derived structural changes at the end of the α E-helix (Supplementary Fig. S4), Akt2 shows a hybrid-like character between Akt1 and Akt3 regarding the SAR of this inhibitor series”.

6. *The authors use molecule 20 as a preferred inhibitor for Akt3, which is not very accurate because the IC50, EC50, cell viability and CETSA stability results all indicate more inhibition by molecule 20 of Akt2 than Akt3.*

We agree with the reviewers' comment and now state that molecule 20 is the most potent molecule for Akt3 from our novel series, which shows equipotent behavior on Akt2. The text now reads: “In agreement with the biochemical data, the aminopyridine (20) and pyridine (21) as well as the methyl pyrazole (12), showed slightly shifted activities towards the isoform myr-Akt3 equipotent to myr-Akt2 underscoring the importance of the nitrogen within the aromatic ring system for targeting Akt3.” Further CETSA experiments were done with higher concentrations of molecule 20 in PANC1 cells (Fig Sl. S16), underlining an equally sufficient stabilization of Akt2 and Akt3.

Minor points:

1. *Figures such as Figure 1c and 1d are low resolution and it's difficult to read the labels below the immunoblot panels.*

We adjusted the labels.

2. *There's a mixture of using “mol/L” and “M” (Molar) in Figure 1c/d and throughout the Methods section, it'd be much clearer if the authors unify them.*

We carefully checked the manuscript to correct for this.

3. *In lines 92-93, the authors mentioned IC50 values of 500 nM, 1nM and 12 nM for Akt3, Akt1 and Akt2, respectively. If these IC50 values are from biochemical assays, the results need to be shown explicitly.*

We thank the referee for this comment and added the according link to table 1.

4. *When the authors describe kinase structures in the manuscript, it would be very helpful if they'd refer to canonical structural features such as the hinge region, the C helix, the DFG motif, the activation loop etc, instead of only referring to residue indices since the numbering system differs from one kinase to another.*

This point is well taken. We changed the description into the c-terminal end of the α E-helix and catalytic loop according to literature known canonical features of kinases.

5. The structural representations throughout this manuscript aren't clear enough. For example: Figure 2b seems a zoom-in of part of Figure 2a, however, the view seems rotated (180 deg?), which can be confusing for readers; in Figure 2C, the "enlarged space in Akt2 and Akt3" needs to be explicitly pointed to; also, in some figures (e.g. Figure 2d, S2A) some explicitly shown residues (and mentioned in the text) are not labeled, which makes it hard to locate them.

We now include more labels and the rotational hint in Fig. 2a. For better visualization, we compiled another figure within the SI (Fig. S4) to highlight the structural differences between the Akt Isoform allosteric pockets.

6. Line 203: "selectivity ratios" should be "selectivity log ratios".

We corrected for this.

REVIEWERS' COMMENTS

Reviewer #2 (Remarks to the Author):

The authors did some of the suggestions although several of the puzzling results were not explained and were deemed beyond the scope of the work.

The following 4 points should be added, corrected and addressed before the acceptance:

1. The authors should elaborate, describe and analyze the changes or lack thereof the newly deposited structures, 7NH4 and 7NH5, in the context of known kinase motif (for example DGF in/out, activation loop, hinge) and in AKT1 specific motifs such as a comparison with linker (aa 136-144) that they observed when compared to structure with the phosphorylated position 473 (PDB ID 6NPZ) or without it (PDB ID 6BUU).

2. Moreover some small changes where not done. For example

The title, 'Akt1 co-crystal structures confirm binding mode' was not changed. Confirmation of what? This should be reworded. AKT1 co-crystal structures displayed inhibitor binding mode. If the structure provides only confirmation, this argues against the value of the study to be published in Nat Comms.

3. All PDBs used should include a reference to their paper

4. Although the authors improved the description of the methods The added methods still are missing details that would prevent their reproducibility, For example, "Cellular studies in a dish in 5 mL of cultivation medium.", the specific medium should be added.

MINOR typos

Page 5; PBD ID should be capitals ; for example 1MRY not 1mry or PDB: 6s9x and thereafter; all PDB should include its reference

Very briefly;, 'very' not needed

'The manual modification of the molecule of the asymmetric unit was performed using the program ...' should be rephrase to the 'manual rebuilding...'

In data availability it says 'Source data are provided with this paper.', which data?

The sentence "Further crystallization studies with full-length Akt2 and Akt3 are pending and will extend our structural understanding and finally provide detailed insights into the interactions of the binding pockets of the three isoforms."

Should be further explained. Do the author means further structural analysis ? do they mean they did it and not included it?

Reviewer #3 (Remarks to the Author):

This authors of this manuscript propose a tool to assess and aid the design of isoform-specific Akt inhibitors, and therefore is potentially of high impact and great interest to a broad group of scientists. The revised the manuscript has addressed all the major and minor concerns I previously raised in a thorough and appropriate manner. Therefore I am happy to recommend this manuscript for publication in Nature Communications.

Response to Decision Letter: Quambusch, Depta, et al.

Manuscript NCOMMS-21-12507A

Reviewer: 2

The authors did some of the suggestions although several of the puzzling results were not explained and were deemed beyond the scope of the work.

The following 4 points should be added, corrected and addressed before the acceptance:

1. The authors should elaborate, describe and analyze the changes or lack thereof the newly deposited structures, 7NH4 and 7NH5, in the context of known kinase motif (for example DGF in/out, activation loop, hinge) and in AKT1 specific motifs such as a comparison with linker (aa 136-144) that they observed when compared to structure with the phosphorylated position 473 (PDB ID 6NPZ) or without it (PDB ID 6BUU).

We thank the referee for this comment. In order to elaborate on the closed conformation and structural changes within the protein, the manuscript now reads: “The complex structures depict a similar binding mode as reported before and show that both ligands stabilize the full-length protein in the autoinhibited conformation with the PH domain folded onto the kinase domain (PH-in conformation). Through this intramolecular contact between the N- and the C-lobe, the regulatory helix α C is displaced, shaping an allosteric binding pocket at the interface. The kinase is stabilized in a DFG-out conformation. Due to missing interactions with adenosine or an equivalent moiety, the hinge region is slightly shifted.”.

Unfortunately, the mentioned linker motif is not resolved (aa 112-145) in our crystal structures due to its high flexibility. Along with this we used a construct lacking the hydrophobic motif (aa 2-446), so we can't give a comparison in that area either.

2. Moreover some small changes where not done. For example

The title, ‘Akt1 co-crystal structures confirm binding mode’ was not changed. Confirmation of what? This should be reworded. AKt1 co-crystal structures displayed inhibitor binding mode. If the structure provides only confirmation, this argues against the value of the study to be published in Nat Comms.

We changed the title to “Akt1 co-crystal structures reveal binding mode.”

3. All PDBs used should include a reference to their paper

We added the specific references.

6S9X ([10.1002/anie.201909857](https://doi.org/10.1002/anie.201909857)), 1MRY ([10.1016/s0969-2126\(02\)00937-1](https://doi.org/10.1016/s0969-2126(02)00937-1)), 6HHF ([10.1158/0008-5472.CAN-18-2861](https://doi.org/10.1158/0008-5472.CAN-18-2861)), 2UVM ([10.1021/cb700019r](https://doi.org/10.1021/cb700019r)), 4GV1 ([10.1021/jm301762v](https://doi.org/10.1021/jm301762v)).

4. Although the authors improved the description of the methods The added methods still are missing details that would prevent their reproducibility, For example, “Cellular studies in a dish in 5 mL of cultivation medium.”, the specific medium should be added.

We revised our methods and added more details to the description.

“...in a dish in 5 mL of cultivation medium (RPMI-1640 (Gibco), 10 % FBS (PAN-Biotech), 1 % penicillin/streptomycin (Gibco)). After 24 h, the transfection mixture (250 µL OptiMEM (Gibco), 2.5 µg pBABE construct, 2.5 µg pCLEco and 7.5 µL TransIT-LT (Mirus) was added to the HEK293T cells. After 72 h of incubation at 37 °C and 5% CO₂ the virus was harvested by centrifugation of the cell suspension at 180 g, virus harvest occurred. The virus-containing supernatant was sterile filtered using a 0.45 µm syringe filter and used directly for infection of Ba/F3 cells or stored at -80 °C.”

We also added a primer list in the supplementary information table 4 and the antibody dilutions in the methods section.

MINOR typos

Page 5; PDB ID should be capitals ; for example 1MRY not 1mry or PDB: 6s9x and thereafter; all PDB should include its reference

We corrected for this.

Very briefly; ‘very’ not needed

Done.

‘The manual modification of the molecule of the asymmetric unit was performed using the program ...’ should be rephrase to the ‘manual rebuilding...’

We rephrased the sentence.

In data availability it says ‘Source data are provided with this paper.’, which data?

We now include a more detailed description of the source data.

The manuscript now reads: “The data supporting the findings of this study are available in the paper and its Supplementary Information. Source Data are provided with this paper. The crystal structure data generated in this study have been deposited in the PDB database under accession code 7NH4 [<http://doi.org/10.2210/pdb7NH4/pdb>] and 7NH5 [<http://doi.org/10.2210/pdb7NH5/pdb>]. Original diffraction data will be made available upon request.”

The sentence “Further crystallization studies with full-length Akt2 and Akt3 are pending and will extend our structural understanding and finally provide detailed insights into the interactions of the binding pockets of the three isoforms.”

Should be further explained. Do the author means further structural analysis ? do they mean they did it and not included it?

This point is well taken. To clarify our statement, we rephrased the sentence and the manuscript now reads: “Further crystallization studies with full-length Akt2 and Akt3 are a necessity to extend our structural understanding and finally provide detailed insights into the interactions of the binding pockets of the three isoforms”.

Reviewer 3:

This authors of this manuscript propose a tool to assess and aid the design of isoform-specific Akt inhibitors, and therefore is potentially of high impact and great interest to a broad group of scientists. The revised the manuscript has addressed all the major and minor concerns I previously raised in a thorough and appropriate manner. Therefore I am happy to recommend this manuscript for publication in Nature Communications.

We are very grateful and thank the reviewer for the enthusiastic assessment of our work.